# A role for MED14 and UVH6 in heterochromatin transcription upon destabilization of silencing

Pierre Bourguet ⓘ, Stève de Bossoreille* ⓘ, Leticia López-González*, Marie-Noëlle Pouch-Pélissier ⓘ, Ángeles Gómez-Zambrano, Anthony Devert, Thierry Pélissier, Romain Pogorelcnik, Isabelle Vaillant, Olivier Mathieu ⓘ

**Constitutive heterochromatin is associated with repressive epigenetic modifications of histones and DNA which silence transcription. Yet, particular mutations or environmental changes can destabilize heterochromatin-associated silencing without noticeable changes in repressive epigenetic marks. Factors allowing transcription in this nonpermissive chromatin context remain poorly known. Here, we show that the transcription factor IIH component UVH6 and the mediator subunit MED14 are both required for heat stress–induced transcriptional changes and release of heterochromatin transcriptional silencing in *Arabidopsis thaliana*. We find that MED14, but not UVH6, is required for transcription when heterochromatin silencing is destabilized in the absence of stress through mutating the MOM1 silencing factor. In this case, our results raise the possibility that transcription dependency over MED14 might require intact patterns of repressive epigenetic marks. We also uncover that MED14 regulates DNA methylation in non-CG contexts at a subset of RNA-directed DNA methylation target loci. These findings provide insight into the control of heterochromatin transcription upon silencing destabilization and identify MED14 as a regulator of DNA methylation.**

## Introduction

In eukaryotic cells, DNA associates with proteins to form chromatin, which is organized in two main states, namely euchromatin and heterochromatin. Compared with the gene-rich euchromatin, heterochromatin is a highly compacted organization of chromatin and mostly comprises different types of repeated sequences, notably transposable elements (TEs). Heterochromatin generally associates with high levels of cytosine DNA methylation and specific histone posttranslational modifications, which in *Arabidopsis thaliana* are dimethylation at histone H3 lysine 9 (H3K9me2) or monomethylation of H3K27 (H3K27me1) (Feng & Michaels, 2015). These epigenetic marks typically contribute to maintaining heterochromatin compaction and transcriptional inactivity. In Arabidopsis, H3K27me1 is catalyzed by the histone methyltransferases *Arabidopsis* trithorax-related protein 5 (ATXR5) and ATXR6 (Jacob et al, 2009). H3K9me2 is deposited by the histone methyltransferases kryptonite (KYP)/SU(VAR)39 homolog 4 (SUVH4), SUVH5, and SUVH6. In plants, cytosine DNA methylation occurs in three nucleotide sequence contexts: CG, CHG, and CHH, where H is any base but a guanine. CG methylation is maintained during DNA replication, where methyltransferase 1 (MET1) reproduces the CG methylation pattern from the template strand to the neosynthetized strand (Law & Jacobsen, 2010). CHG methylation is predominantly mediated by chromomethyltransferase 3 (CMT3), which is recruited to its target sites by binding to H3K9me2 (Du et al, 2012). CHH methylation depends on the activity of both CMT2 and a complex pathway termed RNA-directed DNA methylation (RdDM), which is notably operated by the plant-specific RNA polymerases IV (Pol IV) and V (Pol V) (Law & Jacobsen, 2010; Zemach et al, 2013; Stroud et al, 2014). RdDM relies on small siRNA precursors generated by Pol IV, matured by RNA-dependent RNA polymerase 2 (RDR2) and processed by dicer-like 3 (DCL3) in 24-nucleotide siRNAs that are loaded into argonaute 4 (AGO4) (Matzke & Mosher, 2014). In the canonical model, base pairing of Pol V–dependent scaffold transcripts with AGO4-bound siRNAs recruits domains rearranged methyltransferase 2 (DRM2) to its target sites (Wendte & Pikaard, 2017). Chromatin remodelers also participate in DNA methylation, with defective in RNA-directed DNA methylation 1 (DRD1) promoting CHH methylation at RdDM-dependent loci, whereas decrease in DNA methylation 1 (DDM1) would allow all methyltransferases to access heterochromatin, thereby contributing to DNA methylation in all sequence contexts (Vongs et al, 1993; Kanno et al, 2004; Stroud et al, 2013; Zemach et al, 2013).

Although a certain level of transcription of some heterochromatin sequences is required for establishing or maintaining heterochromatin structure, DNA methylation and histone modifications contribute different layers of silencing that largely repress heterochromatin

Génétique Reproduction et Développement, Centre National de la Recherche Scientifique (CNRS), Inserm, Université Clermont Auvergne, Clermont-Ferrand, France

Correspondence: olivier.mathieu@uca.fr
*Stève de Bossoreille and Leticia López-González contributed equally to this work
Stève de Bossoreille's present address is Laboratoire Reproduction et Développement des Plantes, Université de Lyon, Ecole Normal Supérieure de Lyon, Université Claude Bernard Lyon 1, CNRS, Institut National de la Recherche Agronomique, Lyon, France
Ángeles Gómez-Zambrano's present address is Instituto de Bioquímica Vegetal y Fotosíntesis, Consejo Superior de Investigaciones Científicas-Cartuja, Sevilla, Spain

transcription. Additional factors appear to ensure transcriptional silencing at subsets of heterochromatin loci largely independently of these marks. The best described are Morpheus' molecule 1 (MOM1), replication protein A2 (RPA2), brushy 1 (BRU1), proteins of the Arabidopsis microchidia family (AtMORC), and the maintenance of meristems (MAIN) and MAIN-like 1 (MAIL1) proteins that likely act in complex (Amedeo et al, 2000; Takeda et al, 2004; Elmayan et al, 2005; Kapoor et al, 2005; Moissiard et al, 2012, 2014; Han et al, 2016; Ikeda et al, 2017). Although little is known about the mode of action of these proteins, MAIL1/MAIN and AtMORC6 appear to contribute to heterochromatin compaction, whereas MOM1 does not in spite of its heterochromatic localization (Probst et al, 2003; Feng et al, 2014; Wang et al, 2015; Ikeda et al, 2017).

Some environmental challenges such as heat stress can also transiently alleviate heterochromatin silencing without disturbing epigenetic marks (Lang-Mladek et al, 2010; Pecinka et al, 2010; Tittel-Elmer et al, 2010). Importantly, heat-induced release of silencing does not occur through inhibition of known silencing pathways, nor does it depend on the master regulator of the heat stress transcriptional response HsfA2 (Pecinka et al, 2010; Tittel-Elmer et al, 2010). The H2A.Z histone variant is involved in ambient temperature sensing (Kumar & Wigge, 2010), but its role in heat-induced heterochromatin transcription is unknown. Recent reports suggest that DDM1 and MOM1 act redundantly to re-establish silencing after heat stress, whereas heat-induced expression of the silenced gene *SDC* participates in heat-stress tolerance (Iwasaki & Paszkowski, 2014; Sanchez & Paszkowski, 2014). Interestingly, HIT4 is localized at heterochromatin and is required for its transcription in heat stress but not in the *mom1* mutant (Wang et al, 2013, 2015).

Heterochromatin transcription has been observed in a variety of model organisms under various conditions (Valgardsdottir et al, 2008; Chan & Wong, 2012; Castel & Martienssen, 2013; Saksouk et al, 2015; Negi et al, 2016). Despite its prevalence, heterochromatin transcription is a rather poorly understood process. Notably, how the transcriptional machinery access this repressive chromatin environment remains a largely unsolved question (Feng & Michaels, 2015). To gain insight into this mechanism, we used forward genetics with a reporter-based system and identified the evolutionarily conserved factors XPD/UVH6 and MED14 as required for heterochromatin transcription during heat stress in *A. thaliana*. When heterochromatin silencing is destabilized by mutations in the DDM1- and MOM1-silencing factors, UVH6 is dispensable for transcription, whereas MED14 participates in transcription, specifically in *mom1* mutants. MED14 also targets highly methylated TEs under normal growth conditions, raising the possibility that the repressive chromatin environment might play a role in recruiting MED14. We further show that MED14 regulates non-CG methylation at a subset of loci, likely through RdDM, indicating that MED14 is simultaneously involved in the transcription and the formation of heterochromatin.

# Results

## AtMORC6 and H2A.Z are not involved in release of silencing triggered by heat stress

We and others previously demonstrated that destabilization of silencing by heat stress does not rely on compromising DNA methylation maintenance, RdDM, histone deacetylation, or HsfA2 or MOM1 functions (Lang-Mladek et al, 2010; Pecinka et al, 2010; Tittel-Elmer et al, 2010). We assessed the possible involvement of AtMORC6 and H2A.Z in this process by submitting *atmorc6-3* and *arp6-1* mutants to our previously published heat-stress conditions (Tittel-Elmer et al, 2010). AtMORC6 is required for transcriptional silencing of several repeats and TEs, mostly independently of DNA methylation (Moissiard et al, 2012). ARP6 is involved in assembling H2A.Z-containing nucleosomes, which were shown to be essential to perceiving ambient temperature (Kumar & Wigge, 2010). Reverse transcription followed by quantitative PCR (RT-qPCR) assays at five selected TEs showed transcript over-accumulation under heat stress in WT plants, and this over-accumulation was not significantly affected in *atmorc6-3* and *arp6-1* mutant backgrounds (Fig S1). This suggests that AtMORC6 and deposition of H2A.Z are not necessary for destabilization of silencing induced by heat stress.

## Heat stress–induced release of heterochromatin transcriptional silencing is independent of genome-wide changes in DNA methylation patterns

The Arabidopsis L5 transgenic line contains tandem-repetitions of a transcriptionally silent *β*-glucuronidase (*GUS*) transgene under control of the cauliflower mosaic virus 35S promoter (Morel et al, 2000; Elmayan et al, 2005). Exposing L5 plants to various heat-stress regimes leads to transcriptional de-repression of the *L5-GUS* transgene and numerous endogenous heterochromatin loci (Lang-Mladek et al, 2010; Pecinka et al, 2010; Tittel-Elmer et al, 2010).

We sought to identify the genes required for heat stress–induced activation of heterochromatin transcription using a forward genetics approach, by screening a mutagenized L5 population for reduced *L5-GUS* expression following heat-stress treatment. Because such mutants may potentially be hypersensitive to heat stress and because plants do not survive histochemical detection of GUS accumulation, we set up a screening strategy that consisted in performing GUS staining on isolated leaves from 2-wk-old seedlings after incubation at either 23 or 37°C for 24 h (Fig 1A). Similar to whole seedlings submitted to a 4/37°C temperature shift (Tittel-Elmer et al, 2010), incubating isolated leaves at 37°C led to a robust silencing release of the *L5-GUS* transgene and endogenous repeats and TEs (Fig 1A–C). To further validate the screening method, we defined the impact of this heat stress on gene expression genome-wide, comparing transcriptomes generated by mRNA sequencing (mRNA-seq) in leaves of L5 seedlings (hereafter referred as to WT) following incubation at either 23 or 37°C. Consistent with previous results from ATH1 microarray analysis of whole seedlings exposed to a 4–37°C temperature shift (Tittel-Elmer et al, 2010), we found that regions of constitutive heterochromatin, including centromeric, pericentromeric DNA, and the heterochromatin knob on chromosome 4, were overall transcriptionally activated following incubation at 37°C (Fig 1D). Our mRNA-seq analysis identified a total of 116 up-regulated TEs, mostly located in pericentromeric heterochromatin, confirming that these stress conditions alleviate heterochromatin-associated silencing (Fig 1D). *ONSEN* elements represented notable exceptions among TEs in that they are predominantly located on chromosome arms yet they are highly activated by heat stress (Fig 1D). This is

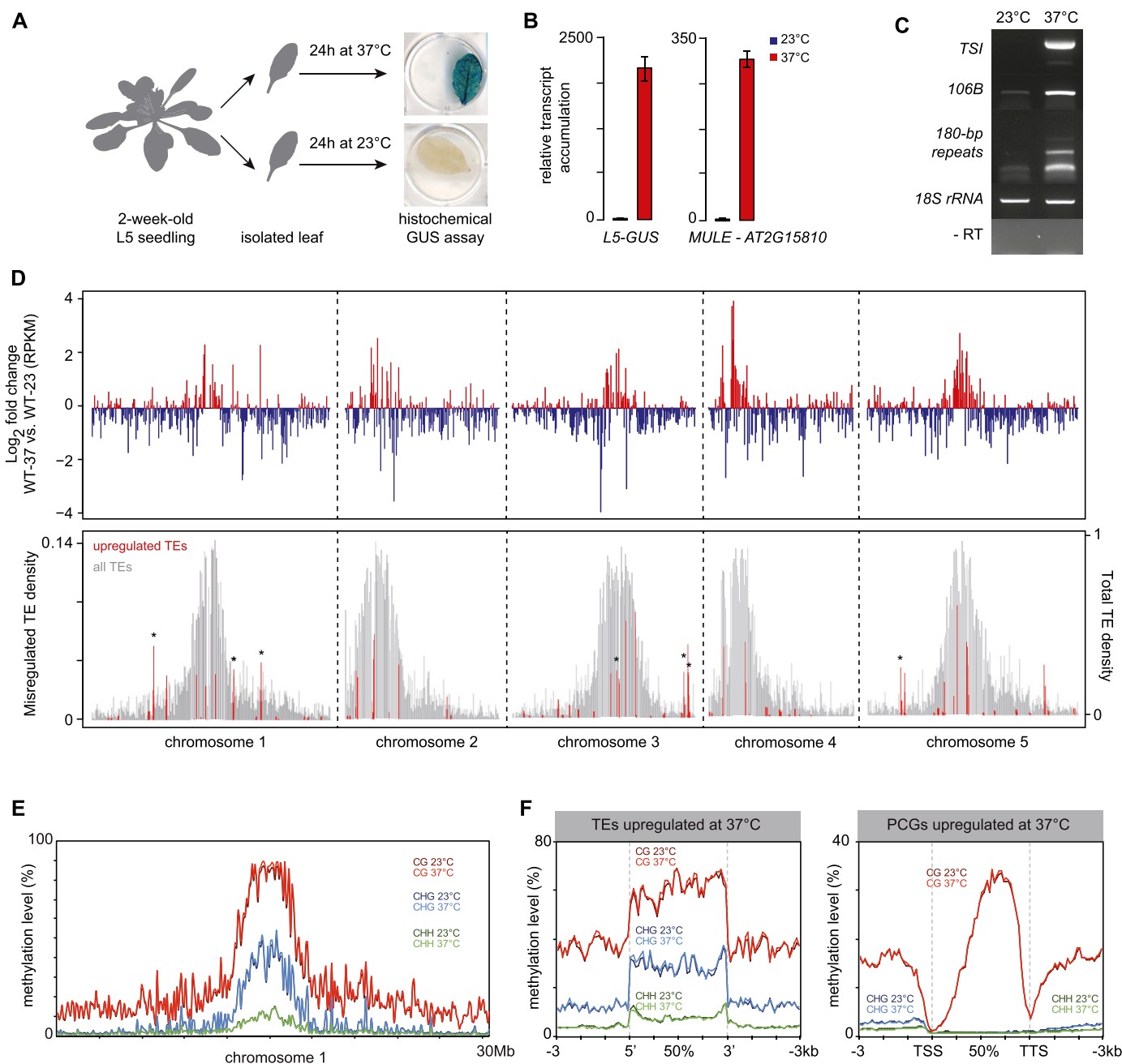

**Figure 1. Heat stress releases heterochromatin silencing without altering DNA methylation.**
**(A)** Scheme representing the method used to submit rosette leaves to a control stress (23°C) or a heat stress (37°C). The *L5-GUS* transgene is reactivated in leaves subjected to heat stress. GUS: β-glucuronidase. **(B)** RT-qPCR analysis of transcripts from *MULE-AT2G15810* and the *L5-GUS* transgene in L5 transgenic plants at 23 or 37°C, normalized to the reference gene *AT5G12240* and further normalized to the mean of L5 samples at 23°C. Error bars represent standard error of the mean across three biological replicates. **(C)** RT-qPCR analysis of transcripts from endogenous repeats in L5 transgenic plants at 23 or 37°C (*Transcriptionally Silent Information* [TSI]). Amplification of 18S rRNA was used as a loading control. PCR in the absence of reverse transcription (RT-) was performed to control for genomic DNA contamination. **(D)** (top) Transcriptional changes in WT plants subjected to heat stress represented along chromosomes by log2 ratios (37/23°C) of mean reads per kilobase per million mapped reads (RPKM) values calculated in 100 kb windows. (bottom) Density of TEs detected as significantly up-regulated in WT plants subjected to heat stress is plotted in red (left y axis) with total TE density in grey (right y axis), both calculated by 100-kb windows. Windows containing up-regulated *ONSEN* elements (*AtCOPIA78*) are marked with an asterisk. **(E)** Average cytosine methylation levels by 500-kb windows calculated in CG, CHG, and CHH contexts in a WT subjected to a control stress (23°C) or to heat stress (37°C). **(F)** PCGs or TEs up-regulated in heat-stressed WT plants were aligned at their 5'-end or 3'-end and average cytosine methylation levels in the indicated nucleotide contexts were calculated from 3 kb upstream to 3 kb downstream in a WT subjected to a control stress (23°C) or to heat stress (37°C). Upstream and downstream regions were divided in 100 bp bins, whereas annotations were divided in 40 bins of equal length.

consistent with previous observations in seedlings exposed to heat stress (Ito et al, 2011; Pecinka et al, 2010; Tittel-Elmer et al, 2010), and occurs owing to the presence of heat-responsive elements in *ONSEN* LTRs (Cavrak et al, 2014). Conversely, transcripts originating from loci located on chromosome arms tended to be down-regulated after 37°C treatment (Fig 1D). Accordingly, protein-coding genes (PCGs) with down-regulated transcript levels were more abundant than the up-regulated ones (4,308 versus 1,487, respectively), a tendency we also previously reported when applying stress on seedlings (Tittel-Elmer et al, 2010). Therefore, applying heat stress to isolated leaves largely mimics the transcriptional response occurring in stressed whole seedlings.

Previous analyses of DNA methylation levels at selected heterochromatin repeats and TEs using methylation-sensitive restriction enzymes have suggested that heat stress–induced alleviation of silencing does not correlate with changes in DNA methylation (Pecinka et al, 2010; Tittel-Elmer et al, 2010). To determine with high resolution whether our heat-stress procedure impacts DNA methylation, we profiled cytosine methylation patterns in WT leaves at 23 and 37°C by whole-genome bisulfite sequencing (BS-seq). Comparison of methylation levels along chromosomes and along all PCGs and TEs revealed no overall impact of heat-stress exposure on DNA methylation (Figs 1E and S2A). Furthermore, TEs and PCGs transcriptionally up-regulated by heat stress displayed similar DNA methylation profiles at 23 and 37°C (Fig 1F). Notably, PCGs up-regulated by heat stress showed higher average WT DNA methylation levels at CG sites than down-regulated PCGs (Fig S2B). Together, these results indicate that heat stress–induced transcriptional changes occur largely independently of detectable variation in DNA methylation patterns.

### Mutants for *UVH6* and *MED14* are deficient for heat stress–induced release of silencing

From the ethyl methanesulfonate (EMS)-mutagenized L5 population, we isolated two mutants that we named *zen1* and *zen2*, where leaves showed reduced GUS staining following incubation at 37°C compared with stressed leaves of the non-mutagenized progenitor L5 line (Fig 2A). RT-qPCR analyses indicated that decreased GUS staining was associated with reduced transcriptional activation of the *L5-GUS* transgene (Fig 2B). Likewise, transcript accumulation from the heterochromatic endogenous loci *TSI*, *106B*, and *MULE* was drastically reduced following heat stress in *zen1* and *zen2* compared with the WT (Fig 2C), demonstrating that suppression of heat stress–mediated release of TGS in *zen1* and *zen2* is not restricted to the *L5* transgene.

The reduced silencing release in stressed *zen* mutants followed a 1:3 (mutant:WT) segregation ratio in F2 populations of *zen1* × L5 and *zen2* × L5 backcrosses, indicating that *zen1* and *zen2* are single, nuclear, recessive mutations. F1 plants from complementation tests between *zen* mutants showed a WT-like response to heat stress, demonstrating that *zen1* and *zen2* mutations affect distinct genes (Fig S3A). Under normal growth conditions, *zen1* plants showed reduced leaf size, altered color, and late flowering, whereas *zen2* seedlings displayed no obvious developmental phenotype (Figs 2D and S3B). Survival assays revealed that both mutants were hypersensitive to heat stress relative to the WT (Fig 2E). We identified

candidate mutations in *zen1* and *zen2* using mapping-by-sequencing from outcross F2 populations (Fig S3C and D). *zen1* plants contained a G to A transition in the *MED14* (AT3G04740) gene, changing tryptophan for a stop codon at amino acid position 1,090 (Fig 2G). We identified a C to T mutation in the *UVH6* (AT1G03190) gene in *zen2* plants, causing a proline to leucine substitution at amino acid 320 (Fig 2G). Complementation of *zen1* and *zen2* phenotypes with transgenes encoding WT versions of MED14 and UVH6 confirmed that the *MED14* and *UVH6* mutations were responsible for the phenotypes observed in *zen1* and *zen2*, respectively (Fig 2F). Hence, *zen1* and *zen2* were renamed *med14-3* and *uvh6-3*, respectively.

MED14 is the central subunit of the MEDIATOR complex, a large protein complex required for early steps of transcription initiation (Cevher et al, 2014; Soutourina, 2018). In Arabidopsis, MED14 function has been involved in cell proliferation and expression regulation of some cold-regulated or biotic stress–induced genes (Autran et al, 2002; Gonzalez et al, 2007; Hemsley et al, 2014; Wang et al, 2016; Zhang et al, 2013). UVH6 is the Arabidopsis ortholog of the human XPD and yeast RAD3 proteins (Liu et al, 2003), which are part of the transcription factor IIH (TFIIH) complex involved in transcription initiation and nucleotide excision repair (Compe & Egly, 2012). XPD is an ATP-dependent 5'-3' helicase and all amino acids required for XPD functions in yeast and humans show remarkable conservation in UVH6 (Kunz et al, 2005). Interestingly, all the mutations identified in *UVH6* disrupt conserved residues (Fig S4). In Arabidopsis, the UVH6 function was first described as necessary for tolerance to UV damage and heat stress (Jenkins et al, 1995, 1997). Failure to isolate homozygous mutants for *uvh6-2*, a transfer-DNA (T-DNA) insertion line, suggested *UVH6* to be an essential gene (Liu et al, 2003). Supporting this conclusion, we also failed to obtain homozygous plants for another *uvh6* T-DNA insertion line (*uvh6-5*) (Fig 2G).

### Transcriptomic analysis of *uvh6* and *med14* mutants in the absence of stress

To investigate the impact of *med14-3* and *uvh6-3* mutations on transcription genome-wide, we determined mRNA profiles of mutant leaves following incubation at either 23°C (*med14-3_23*, *uvh6-3_23*) or 37°C (*med14-3_37*, *uvh6-3_37*) by mRNA-seq. In this analysis, we also profiled the transcriptome at 23°C of another mutant allele of *UVH6* (*uvh6-4*), which we isolated later while pursuing screening our L5 mutant population (Fig S5A and B). The *uvh6-4* mutation replaces a proline for a leucine at amino acid position 532 (Fig 2G). Unlike *uvh6-3*, *uvh6-4* mutants showed yellow-green leaves and reduced stature, a phenotype similar to the one previously described for the *uvh6-1* mutant (Fig S5C) (Jenkins et al, 1997). Suppression of heat stress–induced release of silencing was stronger in *uvh6-4* than in *uvh6-3* (Fig S5B), and survival assays showed that *uvh6-4* and *uvh6-1* plants were more sensitive to heat stress than *uvh6-3* plants (Fig 2E). This indicates that *uvh6-4* is a stronger mutant allele of *UVH6* than *uvh6-3*.

We first compared the mutant transcriptomes with that of the WT in the absence of heat stress. By applying stringent thresholds (fold change ≥ 4, false discovery rate < 0.01), we identified 628 differentially expressed genes (DEGs) in *med14-3_23* (Fig 3A), predominantly PCGs (597). As expected for a mutation of a protein

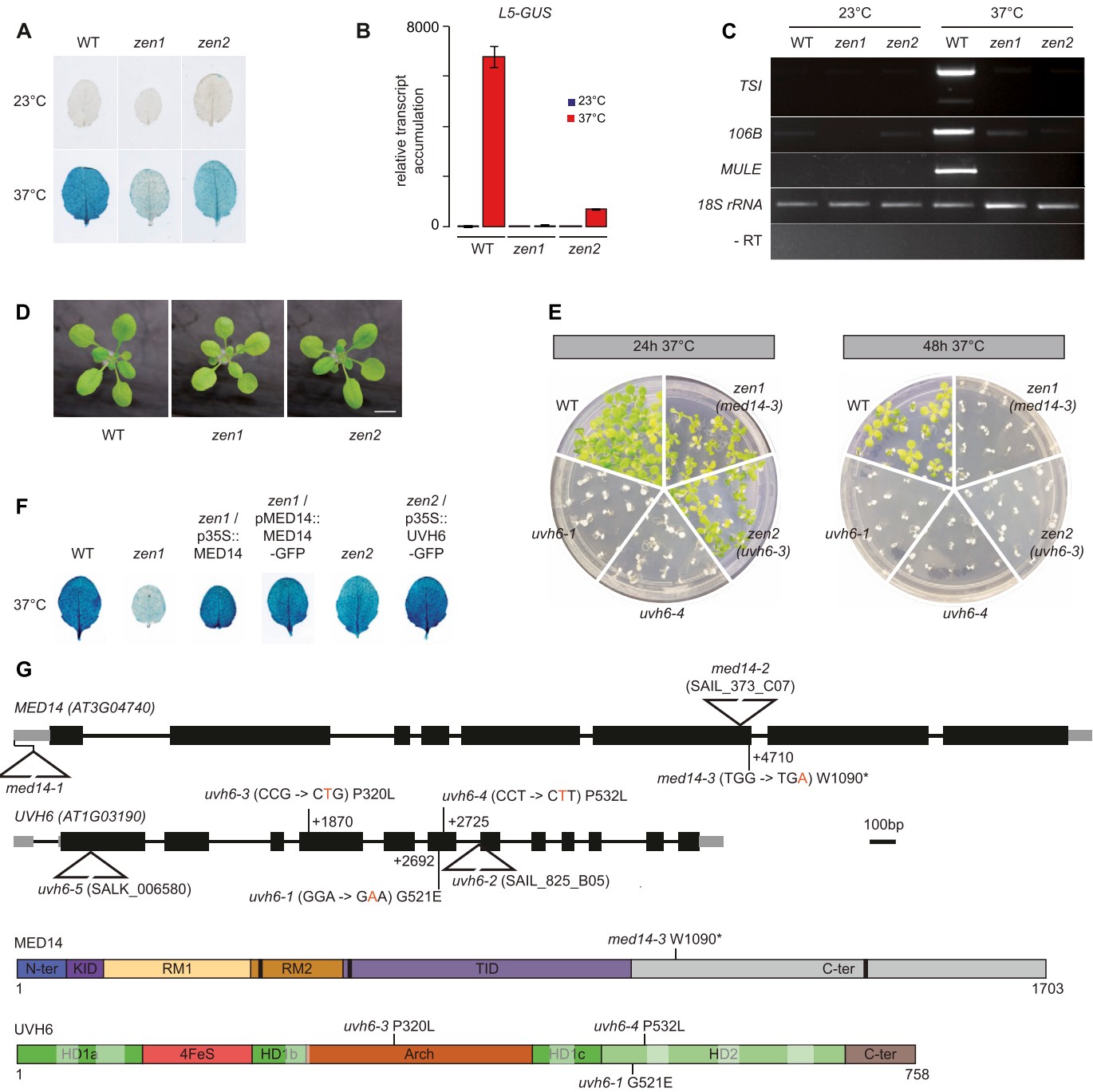

**Figure 2. Mutants for UVH6 and MED14 are impaired in heat stress–induced release of silencing.**
**(A)** Heat stress—induced activation of the *L5-GUS* transgene in rosette leaves of WT (L5 background) and mutants after 24 h at 23°C or 37°C detected by histochemical β-glucuronidase (GUS) staining. **(B)** RT-qPCR analysis of transcripts from the *L5-GUS* transgene in rosette leaves of WT and mutants, normalized to the reference gene *AT5G12240* and further normalized to the mean of WT samples at 23°C. Error bars represent standard error of the mean across three biological replicates. **(C)** RT-PCR analysis of transcripts from endogenous repeats. Amplification of 18S rRNAs was used as a loading control. PCR in the absence of reverse transcription (RT-) was performed to control for genomic DNA contamination. **(D)** Representative pictures of 16-d-old seedlings of the indicated genotypes grown in soil and in long day conditions. Scale bar: 1 cm. **(E)** Heat survival assays. 7-d-old seedlings of the indicated genotypes were subjected to a 37°C heat stress for 24 h or 48 h and returned to standard conditions for 9 d. Pictures are representative of five replicates for the 24 h stress and two replicates for the 48 h stress. **(F)** Heat stress–induced activation of the *L5-GUS* transgene in rosette leaves of the indicated genotypes after 24 h at 37°C detected by GUS staining. **(G)** Top: Gene models for *MED14* and *UVH6*, to scale. Punctual mutations (in orange) and their corresponding amino acid changes are indicated by vertical lines, their position relative to the transcriptional start site (+1) is given. Insertional T-DNA mutations are indicated by triangles. Location of the *med14-1* mutation is reported according to Autran et al (2002). Bottom: Representation of MED14 and UVH6 proteins and their domains. The relative length of MED14 and UVH6 are not to scale. Point mutations and their corresponding amino acid changes are indicated by vertical lines. In MED14, LXXLL motifs have been indicated by black boxes. In UVH6, helicase motifs I, Ia, II, III, IV, V, and VI are indicated by transparent white boxes, respectively, from left to right. The positions of the domains were inferred from studies in other model organisms (see the Materials and Methods section). HD, helicase domain; KID, knob interaction domain; RM, repeat motif; TID, tail interaction domain.

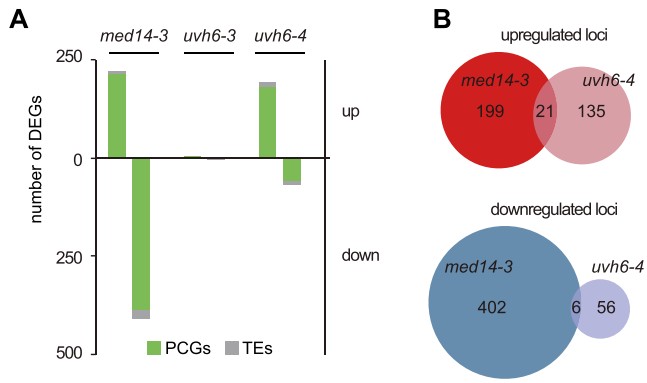

**Figure 3. Transcriptomic analysis of med14 and uvh6 mutants at 23°C.**
**(A)** Number of PCGs and TEs detected as DEGs in *med14-3*, *uvh6-3* and *uvh6-4* relative to the WT at 23°C. **(B)** Venn diagrams showing the extent of the overlap between upregulated and down-regulated loci determined in *med14-3* and *uvh6-4*.

required for transcription, the majority of *med14-3* DEGs (385), including 23 TEs, showed decreased transcript accumulation. Only seven DEGs were detected in *uvh6-3*_23, whereas 218 loci show differential transcript accumulation in *uvh6-4*_23, in agreement with *uvh6-4* being a stronger mutant allele of *UVH6*. Unexpectedly, of the 218 *uvh6-4*_23 DEGs, 156 were up-regulated, suggesting that UVH6 mainly represses transcription at a subset of genomic loci at 23°C (Fig 3A). Loci down-regulated in *uvh6-4* (62) also show reduced transcript accumulation in *uvh6-3* (Fig S6A). The *med14* and *uvh6* mutations affect transcript accumulation at largely independent sets of loci (Fig 3B).

Gene ontology analysis indicated that genes up-regulated in *med14-3*_23 were enriched for biotic stress response genes (Table S1). A similar enrichment was observed in *uvh6-4*_23 up-regulated genes and in genes commonly up-regulated in *med14-3*_23 and *uvh6-4*_23, indicating that MED14 and UVH6 repress genes involved in pathogen response. PCGs down-regulated in *med14-3*_23 were enriched for genes associated with "positive regulation of transcription from RNA polymerase II promoter in response to heat stress". These included *HsfB2A*, *HsfA4A*, *HsfA6b*, and *HsfA3*. *HsfA6b*, *HsfA3*, and another *med14-3*_23 down-regulated gene, *DREB2A*, are partially required for thermotolerance (Huang et al, 2016; Sakuma et al, 2006; Schramm et al, 2007), suggesting that down-regulation of these genes might be responsible for *med14-3* hypersensitivity to heat stress (Fig 2E). PCGs down-regulated in *uvh6-4*_23 were enriched for genes associated with "response to UV" and genes involved in processes such as "anthocyanin biosynthesis", "regulation of flavonoids", "phenylpropanoid metabolism", which protect plants against UV radiation (Jansen et al, 1998). Therefore, downregulation of these genes likely plays a role in *uvh6* mutant UV hypersensitivity (Fig S6B) (Jenkins et al, 1995).

### Genome-wide suppression of heat stress–induced transcriptional activation in *uvh6* and *med14*

To assess the impact of *med14* and *uvh6* on transcript levels following heat stress, we compared *med14-3*_37 and *uvh6-3*_37 with WT-37 mRNA-seq datasets. Overall, heat stress–induced transcriptional activation of pericentromeric sequences was

diminished in *med14* and *uvh6* mutant backgrounds, and transcripts from loci located on chromosome arms tended to accumulate at a lower level than in stressed WT plants (Fig 4A). Compared with *med14*, the impact of the *uvh6* mutation on stress-induced transcriptional changes appeared more global (Figs 4A and S7). Accordingly, the number of DEGs was higher in *uvh6-3*_37 than in *med14-3*_37. We defined 1,631 DEGs in *med14-3*_37, with the vast majority (1,239) showing down-regulation (Fig 4B). Down-regulated loci included 1,124 PCGs and 115 TEs. Although we detected only seven DEGs in *uvh6-3*_23 (Fig 3A), more than 6,200 loci were differentially expressed in *uvh6-3*_37, with 80% of these (4,949) being down-regulated. A total of 4,711 PCGs and 238 TEs displayed less transcript accumulation in *uvh6-3*_37 relative to WT-37 (Fig 4B). The higher number of DEGs at 37°C relative to 23°C in the mutants indicates that MED14 and UVH6 functions are required for efficient transcription of a higher number of loci under heat stress.

PCGs up-regulated by heat stress showed overall reduced transcript levels in *uvh6-3* and *med14-3*, whereas transcript accumulation of PCGs down-regulated by heat stress showed limited changes in *med14-3* compared with *uvh6-3* (Fig S8A and B), suggesting again a more global impact of the *uvh6* mutation on stress-induced transcriptional changes.

In the absence of stress, the *med14* and *uvh6* mutations affect transcript accumulation at rather few, largely independent sets of loci (Fig 3B). Under heat stress, many loci down-regulated in *med14-3* were similarly affected in *uvh6-3* (Fig 4C). Even though this could be expected given the large number of genes down-regulated in *uvh6-3*, we also observed that loci up-regulated in one mutant also showed a similar tendency in the other (Figs 4C and S8C). This is noticeable as in both mutants, up-regulation events are rare relative to down-regulation events. These data suggest that MED14 and UVH6 have converging functions at many overlapping loci under heat-stress conditions.

To try and better understand the molecular circuits through which MED14 and UVH6 modulate gene expression during heat stress, we used published DNA affinity purification sequencing datasets of Arabidopsis transcription factor (TF) binding sites (O'Malley et al, 2016) and determined TF association with PCGs up-regulated by heat stress and PCGs down-regulated in *med14-3*_37 or *uvh6-3*_37 relative to WT-37 (Fig S9A–C). Expectedly, PCGs up-regulated by heat stress were highly enriched for genes associated with binding sites of TFs belonging to the HSF family. A large number of PCGs down-regulated in *med14-3*-37 relative to WT-37 were bound by TFs of the NAC and AP2/EREBP families, known to play important roles in stress response, including response to high temperature and drought stress (Song et al, 2005; Shahnejat-Bushehri et al, 2012; Shao et al, 2015; Obaid et al, 2016; Liu & Zhang, 2017). Comparatively, only a limited number of PCGs down-regulated in *uvh6-3*_37 relative to WT-37 (54 of 216) were significantly enriched for genes containing binding sites of one TF (DREB19). This suggests that MED14 controls expression of rather specific gene sets, whereas UVH6 regulates expression of a broader gene spectrum during heat stress. The mediator complex is thought to directly interact with TFs and bridge them to the transcriptional machinery (Jeronimo & Robert, 2017). Assessing the involvement of the identified TFs in heat stress response will be the subject of further studies.

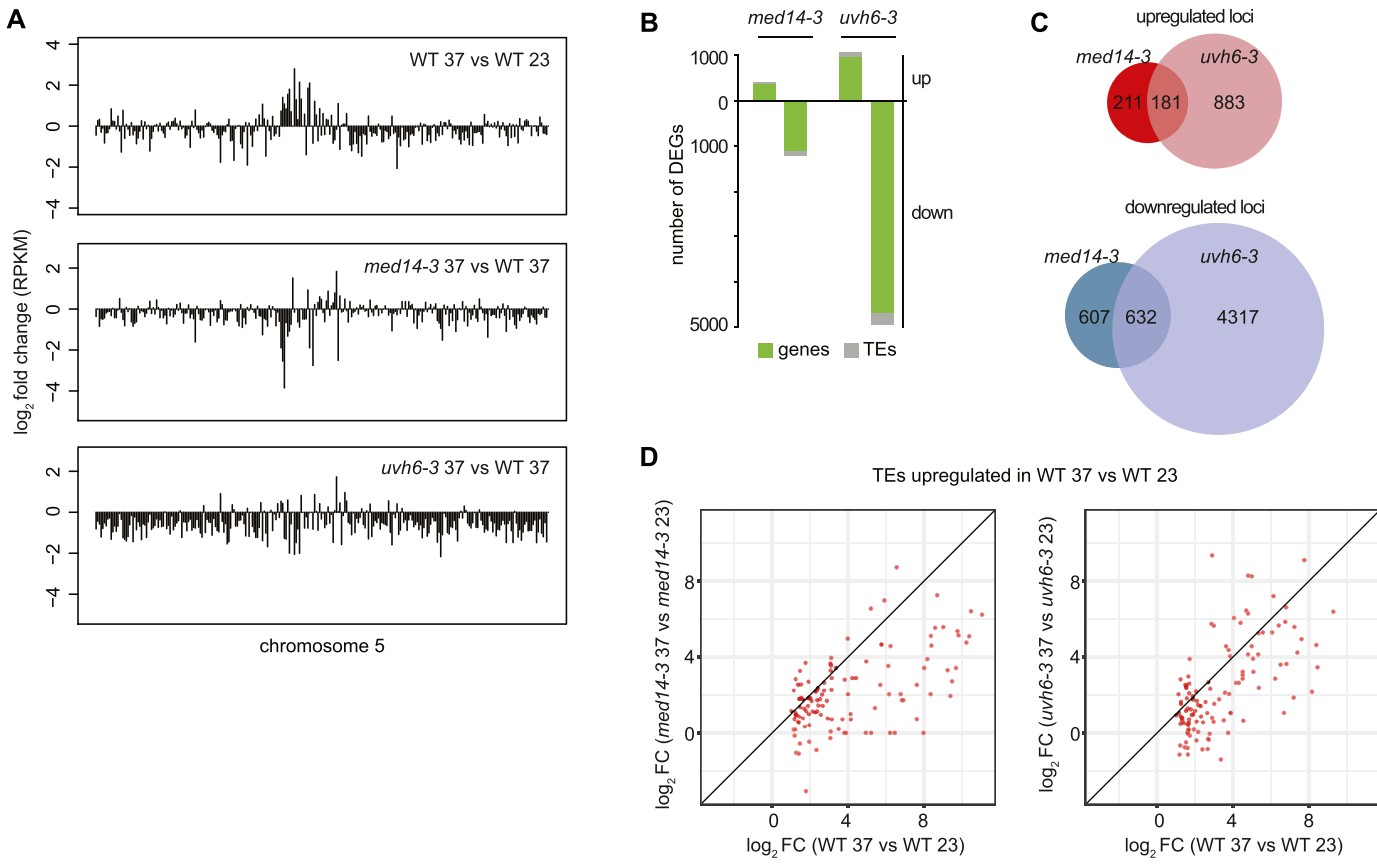

**Figure 4. Transcriptomic analysis of med14 and uvh6 mutants at 37°C.**
**(A)** Transcriptional changes in WT (L5 background) plants subjected to heat stress (top), in *med14-3* at 37°C (middle) and *uvh6-3* at 37°C (bottom) relative to the WT at 37°C, represented along the chromosome five by log$_2$ ratios of mean reads per kilobase per million mapped reads (RPKM) values calculated in 100 kb windows. **(B)** Number of PCGs and TEs detected as DEGs in *med14-3* at 37°C and *uvh6-3* at 37°C relative to the WT at 37°C. **(C)** Venn diagrams showing the extent of the overlap between up-regulated and down-regulated loci determined in *med14-3* at 37°C and *uvh6-3* at 37°C. **(D)** Log$_2$ fold change values at 37°C versus 23°C in *med14-3* (left) and *uvh6-3* (right) plotted against the log$_2$ fold change values at 37°C versus 23°C in the WT (x axis), considering TEs up-regulated in heat-stressed WT plants.

TEs transcriptionally up-regulated by heat stress showed overall reduced transcriptional activation in the mutant backgrounds (Figs 4D and S9D). Heat stress predominantly destabilized silencing at TEs of the DNA/En-Spm, DNA/MuDR, LTR/Copia, and LTR/Gypsy superfamilies (Fig S9E). Among these stress-induced TEs, TEs down-regulated in *uvh6-3*_37 showed comparable proportions. Noticeably, TEs down-regulated in *med14-3*_37 were enriched in LTR/Copia and LTR/Gypsy elements, suggesting that MED14 is preferentially required for heat-induced release of silencing at LTR retrotransposons.

We generated *med14-3 uvh6-3* double mutants and assessed transcript accumulation from *L5-GUS* and selected TEs, including ONSEN, MULE, TSI, VANDAL20 (*AT5TE46155*), and ATCOPIA28 using RT-qPCR (Fig S10). We found no synergy between the two mutations; at a given locus, the transcript levels in *med14-3 uvh6-3* were similar to the ones detected in the mutant showing the strongest down-regulation. These results suggest that, at least at these TEs, MED14 and UVH6, function in the same molecular pathway to promote transcription.

Together, our results indicate that MED14 and UVH6 are required for proper heat stress–induced transcriptional activation of heterochromatic TEs, and more generally play an important role in

controlling transcription at a high number of genomic loci under stress conditions.

## Transcription of methylated TEs requires MED14 but not UVH6

Given that UVH6 and MED14 are involved in transcriptional activation induced by heat-stress, we questioned whether their functions are also required for heterochromatin transcription occurring in mutants for epigenetic regulators. To address this question, we introduced *uvh6-4* and *med14-3* in the *mom1-2* and *ddm1-2* mutant backgrounds, which display constitutive release of transcriptional silencing at heterochromatic loci, and performed mRNA-seq. In *ddm1*, loss of silencing is associated with a strong reduction in DNA, H3K9me2, and H3K27me1 methylation levels (Vongs et al, 1993; Zemach et al, 2013; Ikeda et al, 2017), whereas silencing defects in *mom1* mutants occur without major changes in these epigenetic marks (Amedeo et al, 2000; Habu et al, 2006; Vaillant et al, 2006; Moissiard et al, 2014; Han et al, 2016).

We identified 1909 and 94 TEs significantly up-regulated in *ddm1-2* and *mom1-2*, respectively. Most TEs de-repressed in *mom1-2* overlapped with those de-repressed in the *ddm1-2* mutants (Fig 5A), consistent with MOM1 targeting a subset of methylated TEs for

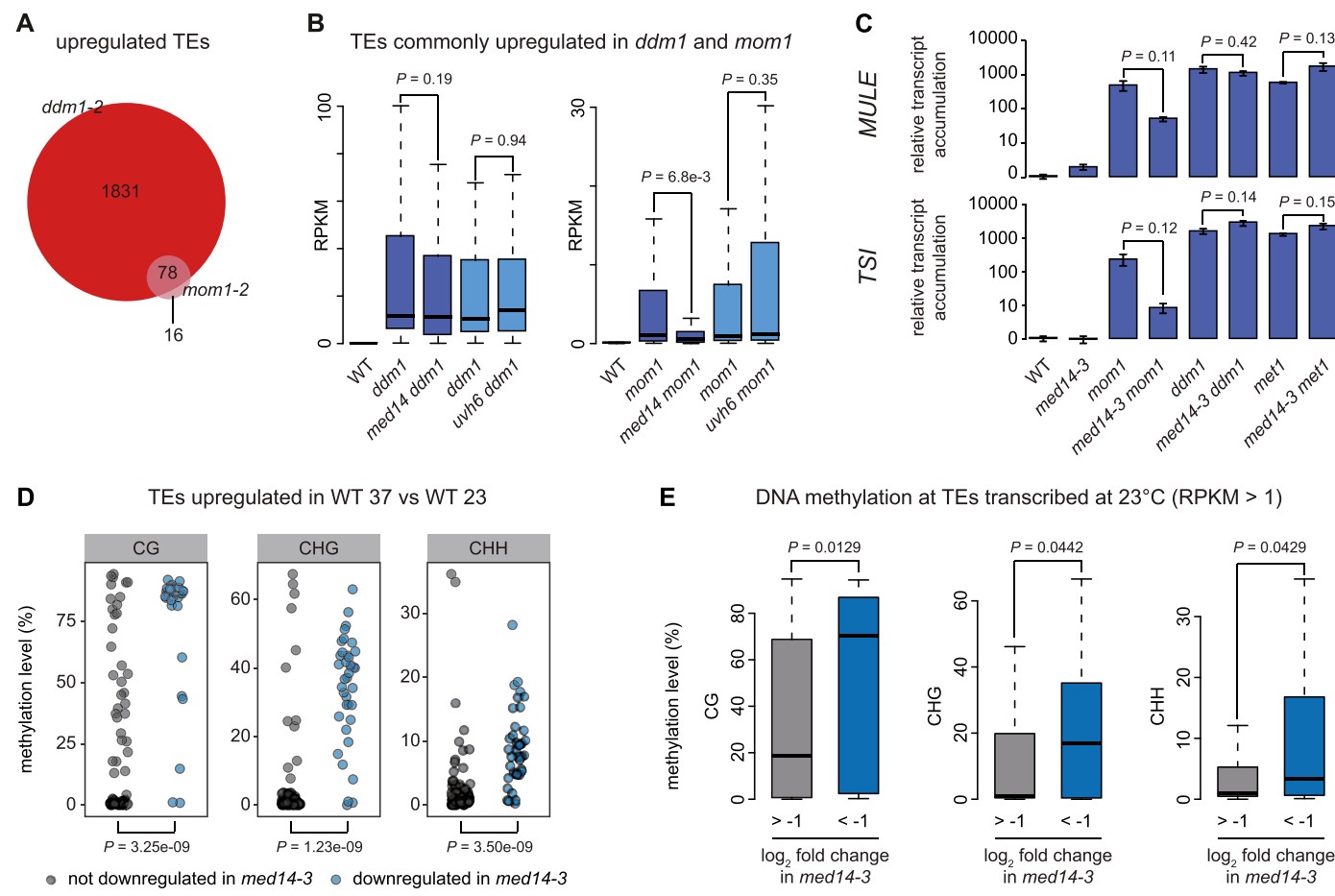

**Figure 5. MED14 promotes transcript accumulation of heterochromatic loci.**
**(A)** Venn diagrams showing the up-regulated TEs in *ddm1* and *mom1* and their overlap. **(B)** Reads per kilobase per million mapped reads (RPKM) values in WT (L5 background) and indicated mutants of TEs commonly up-regulated between *ddm1* and *mom1*. Progenies from sister plants were identically colored. Statistical differences between distributions of single mutants (*ddm1* and *mom1*) versus double mutants (*med14 ddm1*, *uvh6 ddm1*, *med14 mom1*, *uvh6 mom1*) were tested using unpaired two-sided Mann–Whitney tests. **(C)** Transcripts from *TSI* and *MULE* loci were analyzed by RT-qPCR in rosette leaves from indicated genotypes at control temperature (23°C). Data were normalized to the reference gene *AT5G12240* and further normalized to the mean of WT samples at 23°C. Error bars illustrate standard errors of the mean across three biological replicates. Statistically significant differences between means of *mom1*, *ddm1*, *met1*, and combinations of these mutations with *med14-3* were tested using unpaired bilateral *t* tests. **(D)** DNA methylation levels at the CG, CHG, and CHH contexts of TEs up-regulated in heat-stressed WT samples, distinguishing TEs down-regulated in *med14-3* at 37°C from TEs not down-regulated in *med14-3* at 37°C (relative to WT at 37°C), were calculated in WT samples subjected to a control stress at 23°C. Statistical differences between datasets were tested using unpaired two-sided Mann–Whitney tests. **(E)** DNA methylation levels at CG, CHG, and CHH contexts in WT at 23°C were calculated for the indicated groups of TEs. RPKM values at TEs were calculated using multi- and uniquely mapped reads in WT and *med14-3* in control conditions (23°C) (see the Materials and Methods section), and TEs above one RPKM in WT were grouped according to their log$_2$ fold change in *med14-3*. Statistical differences between datasets were tested using unpaired two-sided Mann–Whitney tests.

silencing. Overall, TEs up-regulated in *ddm1-2* accumulated slightly decreased transcript levels in *med14-3 ddm1-2* and weakly increased transcript levels in *uvh6-4 ddm1-2* (Fig S11A). Interestingly, TEs de-repressed by the *mom1-2* mutation showed a strong reduction of transcript levels in *med14-3 mom1-2* (Fig S11B). Although not statistically significant, TE up-regulation tends to be stronger in *uvh6-4 mom1-2*. This suggests a strong dependency over MED14 for TE transcription in *mom1-2*, whereas TE transcription in the *ddm1-2* background is mostly independent of MED14. To strengthen this conclusion, we narrowed down the analysis to the TEs commonly de-repressed by both *ddm1-2* and *mom1-2* mutations. Again, at these 78 TEs, *med14-3*, and *uvh6-4* mutations had no significant impact on *ddm1*-induced release of silencing, whereas transcript levels in *med14-3 mom1-2* were strongly reduced relative to *mom1-*

2 but not in *uvh6-4 mom1-2* (Fig 5B). Because DNA and H3K9me2/K27me1 methylation levels are largely reduced in *ddm1-2*, while being mostly unaltered in *mom1-2* (Fig S11C) (Amedeo et al, 2000; Habu et al, 2006; Vaillant et al, 2006; Moissiard et al, 2014; Han et al, 2016), our data suggest that MED14 is involved in transcription at a subset of heterochromatic TEs when silencing is destabilized without alteration in DDM1-regulated epigenetic marks. Supporting a possible role for DNA methylation in MED14 function, RT-qPCR assays showed that silencing release of *MULE* and *TSI* in the DNA hypomethylated *met1-3* background was not suppressed by the *med14-3* mutation (Fig 5C). Remarkably, when considering TEs up-regulated by heat stress, TEs depending on MED14 for transcriptional up-regulation showed higher DNA methylation levels at all cytosine contexts compared with those independent of the

*med14-3* mutation (Figs 5D and S11D). Such strong bias for highly methylated elements was not observed at TEs that depended on *UVH6* for heat stress–induced transcriptional up-regulation (Fig S11E). Furthermore, TEs transcribed in the WT in the absence of stress and down-regulated by *med14-3* were more methylated than those unaffected by the *med14-3* mutation (Fig 5E), suggesting that MED14 promotes transcript accumulation at a set of highly methylated TEs. On the other hand, UVH6 is required for transcription in a heat stress–specific manner and appears to show a less pronounced preference than MED14 for highly methylated TEs.

### MED14 regulates non-CG DNA methylation

Unlike the vast majority of loci transcriptionally activated in *mom1-2*, the L5-GUS transgene tended to accumulate higher transcript levels in *mom1-2 med14-3* relative to *mom1-2* (Fig S12A). In addition, we identified several endogenous TEs (36) that were transcriptionally de-repressed in *mom1-2 med14-3* plants and were not necessarily activated in either *mom1* or *med14* single mutants (Fig S12B and C). This suggests that although MED14 is largely required for transcription of TEs activated in *mom1-2*, MED14 may also contribute a layer of silencing at some loci. We sought to determine whether the *med14* mutation affects DNA methylation by profiling genome-wide DNA methylation levels in WT and *med14-3* seedlings by BS-seq. Overall, DNA methylation levels were mostly unaltered at CG sites, and showed a moderate reduction at non-CG sites in *med14* compared with the WT (Fig 6A). Calculating average methylation levels along all genomic PCGs, euchromatic TEs, and pericentromeric TEs revealed that non-CG methylation was specifically decreased at pericentromeric TEs in *med14-3* (Fig S13A and B). Because low variations on average methylation levels could mask strong changes at a limited number of loci, we divided the genome in 100-bp bins and determined differentially methylation regions (DMRs) in *med14-3* relative to the WT. This analysis confirmed that the *med14-3* mutation predominantly induced a decrease in DNA methylation at non-CG sites, and preferentially alters methylation of pericentromeric regions of the chromosomes (Fig 6B and C). CHG and CHH hypomethylation occurred concurrently (Fig S13C) indicating that MED14 regulates non-CG methylation at these loci.

The mediator complex is involved in the initiation of Pol II transcription, and Pol II has been reported to be involved in a pathway that regulates DNA methylation (Stroud et al, 2013). Furthermore, at several heterochromatic loci, the mediator promotes Pol II-mediated production of long noncoding scaffold RNAs, which serve to recruit Pol V to these loci (Kim et al, 2011). To assess whether MED14 and Pol II regulate DNA methylation at the same loci, we determined DNA methylation levels of *med14* hypomethylated DMRs in the *nrpb2-3* Pol II mutant allele using previously published data (Zhai et al, 2015). For the vast majority of these genomic regions, DNA methylation levels were unaltered in *nrpb2-3* (Fig S14A), indicating that MED14 regulates DNA methylation largely independently of Pol II.

In the Arabidopsis genome, CHG methylation is mostly mediated by the H3K9me2-directed CMT3 chromomethylase, whereas CHH methylation is maintained by CMT2 and the RdDM pathway at largely distinct genomic regions (Stroud et al, 2014; Zemach et al, 2013). RdDM requires the production of noncoding RNAs by Pol IV

and Pol V, which are eventually required to target and recruit the RdDM effector complex containing the DRM2 *de novo* methyltransferase to its genomic targets (Matzke & Mosher, 2014). We used published data (Stroud et al, 2013) to determine non-CG methylation levels at *med14* non-CG hypomethylated DMRs in mutants for CMT3, CMT2, Pol IV (NRPD1), Pol V (NRPE1), and DRM1/2. *med14* CHG hypomethylated DMRs showed nearly WT methylation levels in *cmt2*, whereas they were largely hypomethylated in *cmt3* (Fig S14B), in agreement with the prominent role of CMT3 over CMT2 in controlling CHG methylation (Stroud et al, 2014). Interestingly, many *med14* CHG hypo DMRs showed a reduced DNA methylation level in the *nrpd1*, *nrpe1*, and *drm1/2* RdDM mutants (Fig S14B). Strikingly, *med14* CHH hypomethylated DMRs showed a strongly reduced DNA methylation level in these RdDM mutants (Fig 6D). This was not merely because of a genome-wide impact of RdDM deficiency on CHH methylation because the same number of randomly selected genomic regions showed much less reduction in CHH methylation in the RdDM mutants (Fig 6E). Conversely, loci with reduced CHH methylation in *drm1/2*, *nrpd1*, or *nrpe1*, all showed lower CHH methylation in *med14-3* (Fig S14C). Together, these results indicate that MED14 regulates non-CG methylation at a subset of loci, likely through RdDM.

## Discussion

Previous studies have shown that heat stress or mutations in certain silencing factors can trigger heterochromatin transcription without modifying levels of repressive epigenetic marks (Amedeo et al, 2000; Lang-Mladek et al, 2010; Pecinka et al, 2010; Tittel-Elmer et al, 2010; Moissiard et al, 2012). That transcription could occur in an otherwise repressive environment suggested that specific mechanisms were involved (Tittel-Elmer et al, 2010). Here, we identified MED14 and UVH6 as critical factors for heterochromatin transcription during heat stress. We showed that UVH6 is dispensable for heterochromatin transcription in silencing mutants such as *mom1* and *ddm1*, whereas MED14 is solely required when heterochromatic marks are not altered. In addition, we showed that MED14 participates in maintenance of DNA methylation at a subset of RdDM-dependent loci.

XPD, the human UVH6 ortholog, is the central subunit of the TFIIH complex, which is crucial for nucleotide exchange repair and is considered a global TF (Compe & Egly, 2016). Our data show that *uvh6* mutations impair transcription of many genes and TEs specifically at elevated temperature. This suggests that UVH6 is not generally required for transcription initiation in Arabidopsis, but is rather involved in a stress-specific transcription mechanism. Previous studies showed that UVH6 belongs to the most essential factors regarding thermotolerance (Jenkins et al, 1997; Larkindale et al, 2005), although the molecular pathway involved is not known. Interestingly, heat-induced accumulation of the canonical heat-responsive factors HSFs and HSPs is independent of UVH6 (Larkindale et al, 2005; Hu et al, 2015), reinforcing the notion that UVH6 is not required for transcription of all genes during heat stress. Human TFIIH has been shown to be involved in selective transcriptional responses to various *stimuli* through

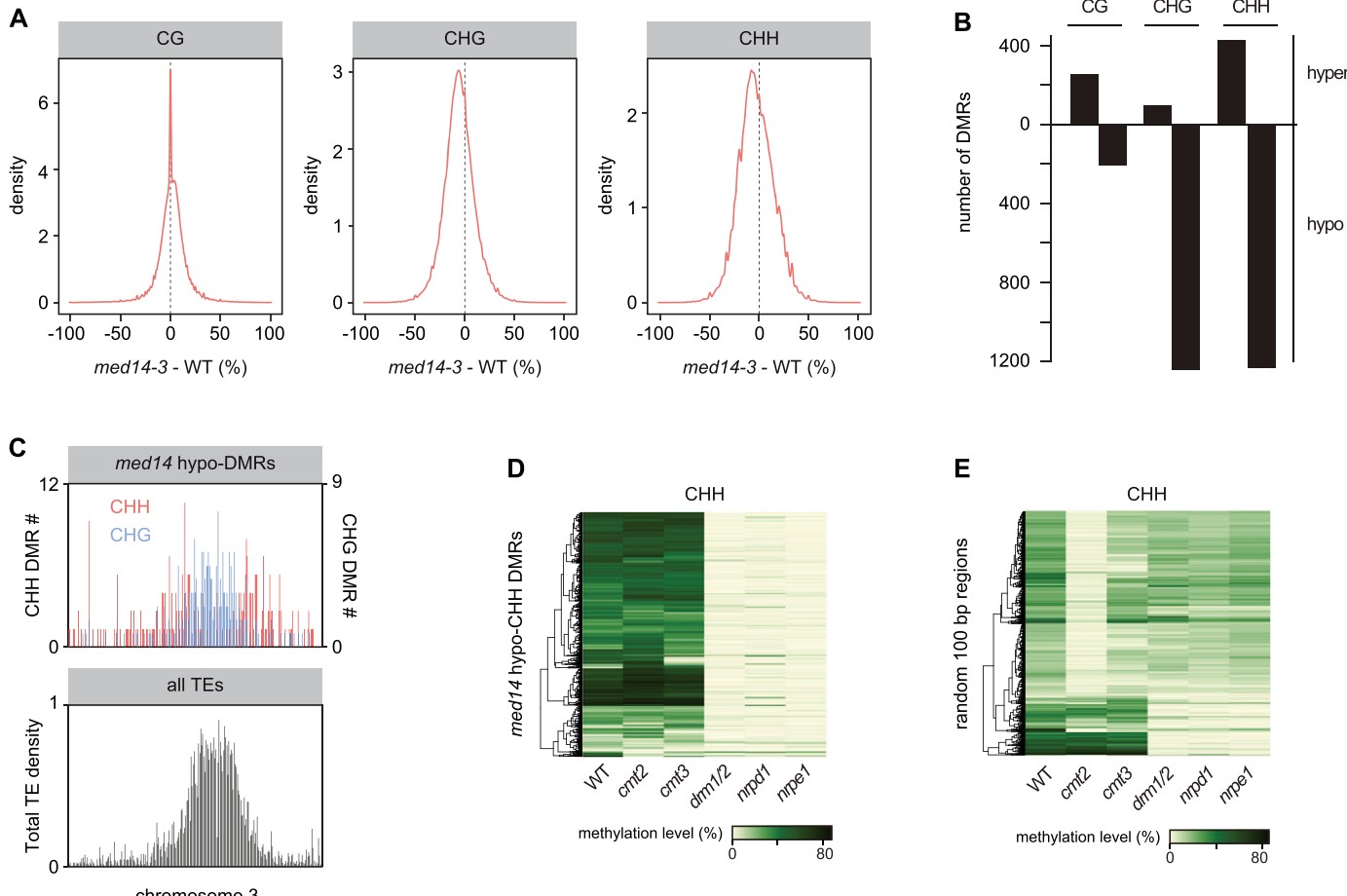

**Figure 6. MED14 controls DNA methylation at CHG and CHH sites.**
**(A)** Kernel density plot of DNA methylation differences between *med14-3* and WT (L5 background) at CG, CHG, and CHH contexts. **(B)** Number of 100-bp differentially methylated regions (DMRs) detected in *med14-3* at CG, CHG, and CHH contexts with a minimum DNA methylation difference of 0.4, 0.2, and 0.2, respectively. **(C)** Chromosomal density of hypo-CHG (blue) and hypo-CHH DMRs (red) identified in *med14-3* (top) with total TE density in grey (bottom), both calculated by 100-kb windows on chromosome 3. **(D)** DNA methylation levels in the CHH context in the indicated genotypes at *med14-3* hypo-CHH DMRs. **(E)** DNA methylation levels in the CHH context in the indicated genotypes at 1,200 randomly selected regions of 100 bp.

posttranslational modifications or recruitment of TFs (Chen et al, 2000; Keriel et al, 2002; Compe et al, 2007; Sano et al, 2007; Chymkowitch et al, 2011; Traboulsi et al, 2014). Therefore, UVH6 may cooperate with HSFs or other TFs during heat stress. In humans, XPD is involved in many functions on top of its well-established roles in transcription and repair, sometimes in a complex other than TFIIH (Compe & Egly, 2016). To get a better understanding of UVH6-dependent transcription in heat stress, future efforts should try to determine whether UVH6 acts as a component of the TFIIH complex or separately.

The mediator is a large protein complex organized in head, middle, and tail modules, with a transiently associated CDK8 kinase module (Soutourina, 2018). MED14 connects the three main modules and is critical for mediator architecture and its function as a co-activator of Pol II transcription (Cevher et al, 2014). We found that MED14 preferentially stimulates transcription of highly methylated TEs in control and stressed conditions. TEs de-repressed in *mom1* mutants require MED14 for transcription, and importantly, the same subset of TEs loose MED14 dependency in the DNA-hypomethylated

*ddm1* background. Similarly, MED14 did not stimulate transcription in a hypomethylated *met1* background. These results raise the possibility that DNA methylation might be required for MED14 targeting to heterochromatin. In yeast, the mediator complex interacts with nucleosomes (Lorch et al, 2000; Zhu et al, 2011a; Liu & Myers, 2012) and the interaction is mediated by histone modifications, although it is not clear how (Zhu et al, 2011b; Uthe et al, 2017). Our data further indicate that MED14 controls DNA methylation at loci where DNA methylation depends on RdDM. RdDM relies on the combined production of non-coding RNAs by the Pol II-related Pol IV and Pol V (Matzke & Mosher, 2014). Compared with transcription initiation by Pol II, less is known about the factors involved in transcription initiation by Pol IV and Pol V. However, epigenetic information also appears crucial for Pol IV and Pol V targeting. Recruitment of Pol IV involves SAWADEE HOMEODOMAIN HOMOLOG 1 (SHH1), a Pol IV-interacting protein that binds to the repressive histone modification H3K9me2 (Law et al, 2013). The SU (VAR)3-9 homologs SUVH2 and SUVH9 are capable of binding methylated DNA and recruit Pol V to DNA methylation (Johnson et al,

2014). Similar to *med14*, *shh1*, and *suvh2/9* mutations also reduce non-CG DNA methylation at a subset RdDM targets. Previous studies proposed that Pol II is required for proper DNA methylation patterns (Stroud et al, 2013) and that the mediator stimulates Pol II–mediated production of non-coding scaffold RNAs that recruits Pol V (Kim et al, 2011). However, we found that DNA methylation at MED14-controlled regions is largely independent of Pol II. Therefore, we propose that MED14 is involved in RdDM at a subset of genomic loci where it might be involved in the early steps of RdDM by acting as a co-activator of Pol IV and/or Pol V.

Although MED14 makes multiple contacts with the different mediator modules, the C-terminal part of yeast and human MED14 has been mapped to the tail module (Tsai et al, 2014; Nozawa et al, 2017). Accordingly, C-terminal truncations of MED14 led to dissociation of the tail module in yeast (Li et al, 1995; Liu & Myers, 2012). The *med14-3* mutation isolated in our study induces a stop codon at amino acid 1,090 of MED14, truncating 614 amino acids at the C-terminal end. The mediator subunits are relatively well conserved between yeast, humans, and Arabidopsis (Bäckström et al, 2007). By analogy, the *med14-3* mutation reported here may be expected to lead to tail-module dissociation. Interestingly, the tail module seems important for recruiting the mediator complex to chromatin (Jeronimo & Robert, 2017; Soutourina, 2018). Thus, the mediator tail module may mediate the preference of MED14 for DNA methylated loci.

In fission yeast, mutations of some subunits from the mediator head and middle modules induce defects in heterochromatin silencing at pericentromeres and concomitant loss of the heterochromatic mark H3K9me2 (Carlsten et al, 2012; Thorsen et al, 2012; Oya et al, 2013). Our transcriptomic data do not support a role for Arabidopsis mediator in heterochromatin silencing. Heterochromatin formation in *Saccharomyces pombe* is dependent on RNAi-dependent and RNAi-independent pathways that both rely on RNA molecules (Martienssen & Moazed, 2015); however, pathways that maintain heterochromatin in Arabidopsis seem largely independent of heterochromatin transcription (Law & Jacobsen, 2010). Therefore, it is possible that mediator stimulates heterochromatin transcription in both model organisms, where it would feed heterochromatin silencing in yeast and RdDM in Arabidopsis. Interestingly, in a yeast mutant background where silencing is compromised, but heterochromatin is maintained, the Med18 mediator subunit is required for heterochromatin transcription of the silent mating-type locus (Oya et al, 2013). This is reminiscent of our observation that MED14 is required for heterochromatin transcription upon heat stress or in the *mom1-2* background, when heterochromatic marks are maintained. Altogether, these findings are consistent with a conserved role of mediator in stimulating heterochromatin transcription.

Heterochromatin transcription, albeit originally counter-intuitive, is a widely reported phenomenon in plants, yeast, drosophila, and mammals. It occurs during specific cell cycles or developmental stages and in stress conditions (Valgardsdottir et al, 2008; Hall et al, 2012; Saksouk et al, 2015; Negi et al, 2016). A well-established function of heterochromatin-derived transcripts is to stimulate heterochromatin formation and/or direct deposition of repressive epigenetic marks (Grewal & Elgin, 2007; Martienssen & Moazed, 2015). Transcripts from heterochromatic regions serve to guide the RdDM pathway in plants, the RNA-induced transcriptional silencing complex in fission yeast (Martienssen & Moazed, 2015), or the piRNA pathway in drosophila (Guzzardo et al, 2013; Andersen et al, 2017). In mammals, the role of heterochromatin transcripts in heterochromatin formation is not clear (Saksouk et al, 2015). Despite its prevalence, the mechanism of heterochromatin transcription remains poorly characterized. Our study uncovers an important role of the conserved proteins XPD/UVH6 and MED14 in this process in Arabidopsis.

# Materials and Methods

## Plant material

The *ddm1-2* (Vongs et al, 1993), *mom1-2* (SAIL_610_G01), *arp6-1* (Kumar & Wigge, 2010) and *atmorc6-3* (Moissiard et al, 2012) mutants are in the Columbia (Col-0) background. The *uvh6-1* mutant is in a Columbia gl1 background (gl1) (Jenkins et al, 1995). The transgenic L5 line was kindly provided by Hervé Vaucheret (Morel et al, 2000). Plants were grown in soil or in vitro in a growth cabinet at 23°C, 50% humidity, using long day conditions (16 h light, 8 h dark). For in vitro conditions, seeds were surface sterilized with calcium hypochlorite and sowed on solid Murashige and Skoog medium containing 1% sucrose (wt/vol). The RNA-seq data for *med14-3* (Figs 3 and 4) was generated with *med14-3* mutants backcrossed five times. For all other molecular data presented in this study, we used lines backcrossed six times for *med14-3* and *uvh6-3* and five times for *uvh6-4*. Point mutations were genotyped by derived Cleaved Amplified Polymorphic Sequence.

## GUS assay

Following heat or control treatment, rosette leaves were transferred to 3 ml of a staining solution composed of 400 μg/ml 5-bromo-4-chloro-3-indolyl-β-D-glucuronic acid, 10 mM EDTA, 50 mM sodium phosphate buffer, pH 7.2, and 0.2% triton X-100. Leaves were placed in a desiccator, subjected to void for 5 min two times, and subsequently incubated 20 h to 24 h at 37°C. Chlorophyll was then repeatedly dissolved in ethanol to allow proper staining visualization.

## Mutagenesis, screening, and mapping

We used EMS-mutagenized seeds from a previously described study (Ikeda et al, 2017). To screen for mutants deficient in heat stress–induced release of silencing of the *L5-GUS* transgene, one leaf per M2 plant was dissected, and leaves from four plants were heat stressed together with 24 h incubation at 37°C in dH$_2$O. Leaves were subsequently subjected to GUS staining as described above. To isolate mutant candidates, a second round of screening was applied to each individual of the M2 pools that contained leaves with reduced GUS signal relative to the non-mutagenized progenitor L5 line.

Mapping-by-sequencing was performed as previously reported (Ikeda et al, 2017). Briefly, we crossed *zen* mutants with *Ler*, selected

F2 segregants with a mutant phenotype (reduced GUS staining after heat stress relative to the L5 line), and bulk-extracted DNA. Libraries were sequenced on an Illumina HiSeq 2,500 instrument at Fasteris SA, generating 100 bp paired-end reads. Sequencing analysis (Ikeda et al, 2017) revealed a locus depleted in genetic markers associated with *Ler*, on chromosome 3 for *med14-3* and on chromosome 1 for *uvh6-3*. Candidate genes with EMS-induced non-synonymous mutations were identified in the mapping interval. Available mutant lines for the candidate genes were analyzed for impaired release of gene silencing upon heat stress, allowing identification of *MED14* and *UVH6*. As indicated in the result section, the *uvh6-4* mutation was identified by complementation test and Sanger sequencing.

### Cloning and complementation

For the pMED14::MED14-GFP construct, the MED14 promoter was PCR amplified from Col-0 genomic DNA from positions −1,311 to −205, where +1 is the adenine of the ATG start codon; the MED14 full-length cDNA was purchased from the plant genome project of RIKEN Genomic Sciences Center (Seki et al, 1998, 2002) and its stop codon was removed by PCR. The promoter and cDNA were cloned into a pBluescript SK plasmid supplemented with attP sites by BP recombination, and subsequently introduced into pB7FWG2 by LR recombination. For the p35S::MED14 construct, the MED14 cDNA was introduced by LR recombination into a pBINHygTX plasmid supplemented with attR sequences. For the p35S::UVH6-GFP construct, the UVH6 cDNA without stop codon was amplified from Col-0 RNA and introduced by BP recombination into the pDONR/ZEO vector (Invitrogen). The fragment was introduced into pH7FWG2 by LR recombination. The *med14-3* and *uvh6-3* mutants were complemented by Agrobacterium-mediated transformation (Clough & Bent, 1998).

### Protein sequence alignments and protein domains

Amino acid sequences were aligned with Clustal Omega v1.2.4. To determine the position of MED14 domains (Fig 2G), *A. thaliana* MED14 was aligned with *S. pombe* MED14 and the domains were determined according to an *S. pombe* structural study (Tsai et al, 2017). The positions of LXXLL motifs (where X is any amino acid), typical of transcriptional co-activators, have been represented for indicative purpose only. *A. thaliana* UVH6 was aligned with *Saccharomyces cerevisiae* RAD3, *Homo sapiens* XPD (Fig S4), and domains were inferred from a joint analysis of RAD3 and XPD (Luo et al, 2015), whereas helicase motif coordinates sourced from a comparative study of eukaryotic and archeal XPD proteins (Wolski et al, 2008).

### Heat stress and UV-C irradiation

Rosette leaves were cut with forceps and transferred to six-well tissue culture plates containing 3 ml dH$_2$O. They were subsequently incubated for 24 h in a 23 or 37°C growth cabinet with otherwise standard conditions. For molecular analysis, 9 to 12 rosette leaves from 3 to 4 seedlings were pooled for heat or control treatment.

Rosette leaves were then dried on absorbent paper, flash-frozen in liquid nitrogen, and stored at −80°C or directly processed.

For survival assays, seeds were sowed in vitro, stratified for 72 h in the dark at 4°C and grown 7 d in standard conditions before heat or UV treatment. Heat stress was applied for 24 h or 48 h. UV-irradiation was performed in an Et-OH sterilized UV chamber (GS Gene Linker; Bio-Rad) equipped with 254 nm bulbs. Plate lids were removed before irradiation at 10,000 J/m$^2$ and placed back immediately. Irradiated seedlings were transferred to a dark growth cabinet with standard conditions for 24 h to block photoreactivation before recovering in light for 5 d.

### RNA analysis

Total RNA was extracted in TRIzol reagent, precipitated with iso-propanol, and washed two times in ethanol 70%. Integrity was assessed by running 1 µg of RNA through an agarose gel after RNA denaturation in 1× MOPS 4% formaldehyde for 15 min at 65°C. 2 µg of RNA were then DNase treated using 2 unit of RQ1 DNAse (Promega) in 15 µl, following manufacturer's instructions. DNase-treated RNAs were further diluted to 40 µl in RNase-free H$_2$O before subsequent analysis. 50 ng of RNA was used as input for reverse transcription PCR (RT-PCR). End-point RT-PCR was performed with the one-step RT-PCR kit (QIAGEN) following manufacturer's instructions in a final volume of 10 µl. For 18S rRNA, MULE, 106B, TSI, and 180 bp, we, respectively, performed 20, 26, 35, 28, and 37 cycles. RT-qPCR was performed in a final volume of 10ul with the SensiFAST SYBR No-ROX One-Step Kit (Bioline) in an Eco Real-time PCR system (Illumina). Quantification cycle (Cq or Ct) values were analyzed following the $2^{-\Delta\Delta CT}$ method (Livak & Schmittgen, 2001). The mean of biological replicates from the control condition was subtracted to each ΔCq value to calculate ΔΔCq. Means and standard errors from biological replicates were calculated from $2^{-\Delta\Delta Cq}$ values.

### RNA-sequencing

Total RNA was extracted and treated as indicated above except that following DNase treatment, RNAs were further purified in phenol-chloroform. Sequencing libraries were generated and sequenced as 50-bp single-end reads at Fasteris SA. Read mapping and quantification of gene expression were performed as previously reported (Ikeda et al, 2017). To allow comparisons between *uvh6-3*, *med14-3*, and *uvh6-4* (Figs 3 and S6), the *uvh6-4* sample and its corresponding WT were artificially converted to non-stranded libraries by merging sense and antisense reads and re-calculating RPKM values at each locus. For comparisons of WT at 37°C versus WT at 23°C, *ddm1-2* versus WT and *mom1-2* versus WT, differentially expressed loci (PCGs and TEs) were defined by a log2-fold change ≥ 1 or ≤ −1, a false discovery rate < 0.01, and only loci defined as differentially expressed in both replicates were retained. When reads could be assigned to a specific strand (*ddm1-2* and *mom1-2* libraries), differential expression was tested in both orientations for each annotation, and only loci that were differentially expressed on the same orientation in both replicates were retained. For all other comparisons, because a single replicate was analyzed, the log2-fold change threshold was increased to ≥ 2 or ≤ −2. Gene ontology analysis was performed using Panther Overrepresentation Test

(December 05, 2017 release) using the December 27, 2017 Gene Ontology database (Ashburner et al, 2000).

To analyze TE transcription in WT and *med14-3* in standard conditions (23°C) (Fig 5E), we aligned reads from WT and *med14-3* with STAR (Dobin et al, 2013) and retained randomly assigned multi-mapped reads. We counted reads on TAIR10 transposon annotations and selected TEs with a minimum RPKM value of one in WT, a minimum length of 200 bp, and that had at most 10% of their length intersecting a PCG annotation, regardless of their orientation.

For transcriptomic studies of *med14-3* in the *ddm1-2* background, we compared *med14-3 ddm1-2* double mutants with *ddm1-2* mutants, both isolated from the F2 progeny of a *med14-3/+ ddm1-2/+* F1 plant. We followed the same method for *uvh6-4 ddm1-2*. For *med14-3* in the *mom1-2* background, we compared *med14-3 mom1-2* double mutants with *mom1-2* mutants, both isolated from the F3 progeny of a *med14-3/+ mom1-2* F2 plant, and followed the same method for *uvh6-4 mom1-2*.

## Whole-genome BS-seq

After 24 h incubation at 23°C in dH2O of 16-d-old rosette leaves from L5 and *med14-3*, genomic DNA was extracted using the Wizard Genomic DNA Purification Kit (Promega) following manufacturer's instructions. One microgram of DNA was used for bisulfite treatment, library preparation, and sequencing on a Hiseq2000 at the Beijing Genomics Institute (Shenzhen, China), producing paired 91-bp oriented reads. We used and re-analyzed previously published BS-seq datasets for *ddm1-2* (GSM981009), *cmt2-7* (GSM981002), *cmt3-11* (GSM981003), *drm1/2* (*drm1-2 drm2-2*; GSM981015), *nrpd1a-4* (GSM981039), *nrpe1-11* (GSM981040), and WT (GSM980986) (Stroud et al, 2013); *mom1-2* (GSM1375964) and WT (GSM1375966) (Moissiard et al, 2014); *nrpb2-3* (GSM1848705, GSM1848706) and WT (GSM1848703, GSM1848704) (Zhai et al, 2015).

PCR duplicates were removed using a custom program: a read pair was considered duplicated if both reads from a pair were identical to both reads of another read pair. We utilized BS-Seeker2 v2.1.5 (Guo et al, 2013) to map libraries on the TAIR10 reference genome using the Bowtie2 aligner with 4% mismatches and call methylation values from uniquely mapped reads. 100-kb window average methylation levels and metaplots of average methylation levels over PCGs or subgroups of TEs were generated in CG, CHG, and CHH contexts with CGmapTools v0.1.0 (Guo et al, 2018). For metaplots, regions of interest were divided in 40 bins of equal length while upstream, and downstream regions extended for 30 bins of 100 base pairs.

DMRs were calculated as previously reported (Stroud et al, 2013), except that contiguous DMRs were not merged, and that the thresholds for minimum methylation differences were 0.4, 0.2, and 0.2 for the CG, CHG, and CHH contexts, respectively. To extract methylation levels at specific regions (e.g., Figs 5D and 6D), we first calculated the methylation level of individual cytosines in the region and extracted the average. For all calculations of methylation levels or DMRs, only cytosines with a minimum coverage of six reads were considered. Sequencing data generated in this study have been deposited to ArrayExpress under accession number E-MTAB-7203 (https://www.ebi.ac.uk/arrayexpress/).

## Statistical analysis

Means and standard errors of the means were calculated from independent biological samples. All analyses were conducted with R version 3.4.0 (R Core Team, 2017). All boxplots had whiskers extend to the furthest data point that was less than 1.5-fold interquartile range from the box (Tukey's definition). Heat maps were generated using the *heatmap.2* function of the *gplots* package with euclidean distance, complete clustering, and without scaling (Warnes et al, 2005). Differences in mean for RT-qPCR data were tested using an unpaired *t* test with Welch's correction with the *t* test function. For RT-qPCR data in Fig S10 because of the interaction between the temperature treatments and genotypes, the data were split between 23 and 37°C. Subsequent analysis of variance was performed with the *aov* function, and *post hoc* analysis was performed with Tukey's honest significant difference test, using the *TukeyHSD* function with a 95% confidence level. Because the strong absolute variance of WT at 37°C prohibited the assessment of differences between mutants, this sample was excluded. Differences in distributions of RPKM values (Figs 5B, S6A, 11A, and B) and methylation values (Figs 5D, E, and S13C) were tested with an unpaired two-sided Mann–Whitney test using the *wilcox test* function.

# Supplementary Information

# Acknowledgments

We are grateful to Korbinian Schneeberger for his help with mapping-by-sequencing and to Jerzy Paszkowski for his support at the early steps of this project. This work was supported by CNRS, Inserm, Université Clermont Auvergne, Young Researcher grants from the Auvergne Regional Council (to I Vaillant and to O Mathieu), an European Molecular Biology Organization Young Investigator award (to O Mathieu), and a grant from the European Research Council (I2ST 260742 to O Mathieu). P Bourguet was supported by a PhD studentship from the Ministère de l'éducation nationale, de l'enseignement supérieur et de la recherche.

## Author Contributions

P Bourguet: conceptualization, data curation, formal analysis, validation, investigation, visualization, methodology, writing—original draft, review, and editing.
S de Bossoreille: conceptualization, formal analysis, validation, investigation, visualization, methodology, and writing—original draft.
L López-González: conceptualization, validation, investigation, methodology, and writing—original draft.
M-N Pouch-Pélissier: investigation and methodology.
Á Gómez-Zambrano: investigation.
A Devert: investigation.
T Pélissier: validation, investigation, and methodology.
R Pogorelcnik: data curation, formal analysis, and methodology.
I Vaillant: conceptualization, funding acquisition, investigation, methodology, and writing—review and editing.
O Mathieu: conceptualization, data curation, formal analysis, supervision, funding acquisition, validation, investigation, visualization, methodology, project administration, and writing—original draft, review, and editing.

**Conflict of Interest Statement**

The authors declare that they have no conflict of interest.

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
