## [Reviewer comments · Life Science Alliance]

Life Science Alliance

A role for MED14 and UVH6 in heterochromatin transcription upon destabilization of silencing

Pierre Bourguet, Stève de Bossoreille, Leticia López-González, Marie-Noëlle Pouch-Pélissier, Ángeles Gómez-Zambrano, Anthony Devert, Thierry Pélissier, Romain Pogorelnik, Isabelle Vaillant, and Olivier Mathieu

DOI: [10.26508/lsa.201800197](https://doi.org/10.26508/lsa.201800197)

Corresponding author(s): Olivier Mathieu, Génétique Reproduction et Développement (GReD), CNRS, Inserm, Université Clermont Auvergne

Review Timeline:	Submission Date:	2018-09-19
	Editorial Decision:	2018-10-24
	Revision Received:	2018-12-03
	Editorial Decision:	2018-12-04
	Revision Received:	2018-12-05
	Accepted:	2018-12-05

Scientific Editor: Andrea Leibfried

Transaction Report:

October 24, 2018

Re: Life Science Alliance manuscript #LSA-2018-00197

Olivier Mathieu

Universite Clermont Auvergne, CNRS, Inserm, Genetique Reproduction et Developpement (GReD);

Dear Dr. Mathieu,

Thank you for submitting your manuscript entitled "Differential requirement of MED14 and UVH6 for heterochromatin transcription upon destabilization of silencing" to Life Science Alliance. The manuscript was assessed by expert reviewers, whose comments are appended to this letter.

As you will see, the reviewers appreciate your data and provide constructive input on how to clarify some aspects and how to further strengthen your work. We would thus like to invite you to revise your work, addressing the points raised by the reviewers. The revision seems minor and straightforward, but please do get in touch in case you would like to discuss individual points further.

Thank you for this interesting contribution to Life Science Alliance. We are looking forward to receiving your revised manuscript.

Sincerely,

Andrea Leibfried, PhD

Executive Editor

Life Science Alliance

Meyerhofstr. 1

69117 Heidelberg, Germany

t +49 6221 8891 502
e a.leibfried@life-science-alliance.org
www.life-science-alliance.org

- A letter addressing the reviewers' comments point by point.
- An editable version of the final text (.DOC or .DOCX) is needed for copyediting (no PDFs).
- High-resolution figure, supplementary figure and video files uploaded as individual files: See our detailed guidelines for preparing your production-ready images, <http://life-science-alliance.org/authorguide>
- Summary blurb (enter in submission system): A short text summarizing in a single sentence the study (max. 200 characters including spaces). This text is used in conjunction with the titles of papers, hence should be informative and complementary to the title and running title. It should describe the context and significance of the findings for a general readership; it should be written in the present tense and refer to the work in the third person. Author names should not be mentioned.

B. MANUSCRIPT ORGANIZATION AND FORMATTING:

Full guidelines are available on our Instructions for Authors page, <http://life-science-alliance.org/authorguide>

Reviewer #1 (Comments to the Authors (Required)):

The authors identified mutations in MED14 and UVH6 in a screen for mutants that do not upregulate a reporter gene upon heat stress. Through Figures 1-4 and the related supplementary figures, they clearly demonstrate that MED14 and UVH6 are required for release of silencing during heat stress. Further analysis strongly demonstrated that MED14 and UVH6 function differently in the silencing release upon heat stress. However, the additional analysis about the function of MED14 in transcription when silencing is destabilized in the absence of stress does not support their conclusion that MED14 is required for transcription of heterochromatic TEs that were modified by DDM1-mediated epigenetic marks.

1. The presentation of results and conclusions surrounding TE expression and *ddm1 med14* or *mom1 med14* double mutants are confusing. In supplementary figure 10A and Figure 5B, upregulated TEs in *ddm1* were not affected by the *med14* mutation, demonstrating that in the heterochromatic regions destabilized because of loss of DDM1-mediated DNA methylation, MED14 is not required for transcription. On the other hand, as the authors pointed out, transcription of the heterochromatic regions destabilized by *mom1* mutation clearly requires MED14 (supplementary figure 10B and Figure 5B). As the author stated, "DNA and H3K9me2/K27me1 methylation levels are largely reduced in *ddm1-2*, while being mostly unaltered in *mom1-2*". Together, these lead to the conclusion that MED14 is required for transcription of a small subset of TEs which are destabilized by MOM1, suggesting that MED14-mediated transcription does not require DNA methylation mediated by DDM1. The conclusion that "MED14 is involved in transcription at a subset of heterochromatic TEs and requires DDM1-mediated epigenetic marks for its function" is not supported.
2. The authors' conclusion that "MED14 promotes transcript accumulation at a set of highly methylated TEs" based on Figure 5D, supplementary figure 10D and Figure 5E is valid. However, there is no evidence supporting the following conclusion: "requires proper DNA methylation patterns for this function".
3. For heat stress-induced gene expression, the authors only compared mutants 37 to WT 37. What about the fold change between 37 and 23 in each mutant compared to the fold change in WT? Specifically, what is the fold change between 37 and 23 in mutants and WT are plotted as Figure 4D?

Minor Comments:

1. In the text of the results it is stated that *zen1* mutants have reduced leaf size, altered color, and late flowering. This is not obvious from the image in Figure 2D. If the authors have data on these traits it should be included in a supplemental figure.
2. In the results section, the use of the word "remarkably" as a preface for stating observations should be reduced.

Reviewer #2 (Comments to the Authors (Required)):

This manuscript identifies MED14 and UVH6 as important components for the transcription of heterochromatin under certain circumstances. Indeed, heterochromatin is generally methylated and silenced, but it can be transcribed under certain stresses, for example heat stress, without removing silencing marks. Using an elegant forward genetic screen, the authors identified *med14* and *uvh6* mutants in which the reactivation of a silenced transgene by heat stress is compromised. Through whole-genome analysis, they show that mostly transposable elements (TEs) of the Copia and Gypsy families are regulated by MED14, whereas UVH6 has a more narrowed effect. Then, by combining *med14* and *uvh6* with *ddm1* and *mom1* mutations, which allow reactivation of TEs with or without demethylation, they show that MED14 is required for the transcription of methylated TEs. The amount of work is impressive and of very high quality. The conclusions are more advanced for MED14 than UVH6, but studying these two mutants provides a good balance. The manuscript is very well organized and suitable for publication. Addressing the following questions would improve the manuscript:

Given that *mom1* has a weaker effect than *ddm1* on the L5 transgene, it would be interesting to look if L5 is more reactivated in the *mom1 med14* double mutant.

Given that, like *mom1*, the *morc* and *main* mutants allow reactivating TEs without affecting methylation, it would be interesting to determine if their effect is also blocked by *med14*.

Minor comments:

In the abstract, the sentence "MED14 is required for transcription when heterochromatin silencing is destabilized in the absence of stress" is unclear. How heterochromatin silencing is destabilized is explained in the text, but it should be clearly said in abstract.

Reviewer #3 (Comments to the Authors (Required)):

Heterochromatin provides a means to tightly pack chromatin and shutdown transcription over relatively large genomic distances. This is particularly important in pericentromeric regions. Despite this large-scale reduction of transcription, as these authors show, heterochromatin can still be activated under stress conditions, despite repressive chromatin marks. In this study the authors have devised a very elegant forward genetic screen to identify key components necessary for transcriptional activation of heterochromatin in detached leaves. The authors go on to clone the underlying genes and characterize them in some detail. The authors show very convincingly that MED14 and UVH6 are key players in this pathway, and likely act together to activate the transcription of many silenced genes in response to heat stress.

This manuscript is extremely well written and very well presented. A number of interesting and well-designed experiments are described and executed and the conclusions drawn are balanced and very well supported. I believe this is a complete study. I see no major gaps or shortcomings. The following are essentially minor suggestions that might help clarify some aspects of the study.

(1) The authors exclude H2A.Z, MORC6 and HsfA2 as being major actors in releasing transcription in silenced chromatin. It would be interesting to know if they have considered HsfA1a class transcription factors. These are upstream of HsfA2 and responsible for the initial response to heat. There are for example several published ChIP-seq datasets for HsfA1 class TFs, so it would be possible to see if these are enriched in the UVH6 and MED14 dependent genes. (Related to this, the authors mention ONSEN, which has HSEs, but it isn't clear if ONSEN is altered in their mutants.

(2) Related to (1) the publicly available DAP-seq data from the Ecker lab might be a useful resource to determine the overall mechanism by which these genes are regulated. Briefly, while heterochromatin is a broad mechanism to shutdown the expression of hundreds or thousands of genes, the specific up-regulation of many of these genes by heat, and the even more specific requirement of a subset of these for UVH6 and MED14 suggests the mechanism is quite specific, suggesting there might be a specific TF responsible. This may well be beyond the scope of this study, but it might be easy to identify from data already available, or it might be interesting if the authors briefly touched on this issue.

(3) Typos: P3: Notably, how the transcriptional machinery can access to a repressive chromatin environment. And: heading: Transcriptomic analysis of *uhv6* and *med14* mutants in the absence of stress

Reviewer #1 (Comments to the Authors (Required)):

The authors identified mutations in MED14 and UVH6 in a screen for mutants that do not upregulate a reporter gene upon heat stress. Through Figures 1-4 and the related supplementary figures, they clearly demonstrate that MED14 and UVH6 are required for release of silencing during heat stress. Further analysis strongly demonstrated that MED14 and UVH6 function differently in the silencing release upon heat stress. However, the additional analysis about the function of MED14 in transcription when silencing is destabilized in the absence of stress does not support their conclusion that MED14 is required for transcription of heterochromatic TEs that were modified by DDM1-mediated epigenetic marks.

1. The presentation of results and conclusions surrounding TE expression and *ddm1 med14* or *mom1 med14* double mutants are confusing. In supplementary figure 10A and Figure 5B, upregulated TEs in *ddm1* were not affected by the *med14* mutation, demonstrating that in the heterochromatic regions destabilized because of loss of DDM1-mediated DNA methylation, MED14 is not required for transcription. On the other hand, as the authors pointed out, transcription of the heterochromatic regions destabilized by *mom1* mutation clearly requires MED14 (supplementary figure 10B and Figure 5B). As the author stated, "DNA and H3K9me2/K27me1 methylation levels are largely reduced in *ddm1-2*, while being mostly unaltered in *mom1-2*". Together, these lead to the conclusion that MED14 is required for transcription of a small subset of TEs which are destabilized by MOM1, suggesting that MED14-mediated transcription does not require DNA methylation mediated by DDM1. The conclusion that "MED14 is involved in transcription at a subset of heterochromatic TEs and requires DDM1-mediated epigenetic marks for its function" is not supported.

We respectfully disagree with the conclusion "MED14-mediated transcription does not require DNA methylation mediated by DDM1" that the reviewer draws from our data.

We show that *ddm1*-induced transcriptional activation of TEs occurs concomitantly with a strong reduction in DNA methylation, and in this case, transcription does not involve MED14 (Fig 5B, C and Fig S11A, C). However, we show that MED14 is required for transcription of the same set of TEs when silencing is destabilized in the *mom1* mutant background (Fig 5B, C and Fig S11A), without alteration in patterns / levels of DDM1-mediated epigenetic marks (Fig S11C and literature). Thus, this suggests that MED14 is involved in transcription of these heterochromatic TEs when silencing is destabilized without alteration in epigenetic marks regulated by DDM1, i.e. in presence of wild-type DNA methylation levels. To try and better convey our conclusion, we have reworded the corresponding sentence which now reads "our data suggest that MED14 is involved in transcription at a subset of heterochromatic TEs when silencing is destabilized without alteration in DDM1-regulated epigenetic marks."

2. The authors' conclusion that "MED14 promotes transcript accumulation at a set of highly methylated TEs" based on Figure 5D, supplementary figure 10D and Figure 5E is valid. However, there is no evidence supporting the following conclusion: "requires proper DNA methylation patterns for this function".

Although we believe our data strongly suggest a connection between MED14-dependent transcription and DNA methylation, we do agree that we do not provide formal proof that DNA methylation is required for MED14 function. This statement has been removed in the revised manuscript.

3. For heat stress-induced gene expression, the authors only compared mutants 37 to WT 37. What about the fold change between 37 and 23 in each mutant compared to the fold change in WT? Specifically, what is the fold change between 37 and 23 in mutants and WT are plotted as Figure 4D?

As suggested, we have calculated the fold change between 37°C and 23°C in mutants and WT at transposable elements upregulated by heat-stress. In agreement with our previous conclusions, the data show reduced transcriptional activation of transposable elements in the *med14-3* and *uvh6-3* mutant backgrounds. The new plots are shown in the revised figure 4D.

Minor Comments:

1. In the text of the results it is stated that *zen1* mutants have reduced leaf size, altered color, and late flowering. This is not obvious from the image in Figure 2D. If the authors have data on these traits it should be included in a supplemental figure.

We have included data on *zen1* late flowering phenotype and new plant pictures in Supplementary Figure 3.

2. In the results section, the use of the word "remarkably" as a preface for stating observations should be reduced.

Done.

Reviewer #2 (Comments to the Authors (Required)):

This manuscript identifies MED14 and UVH6 as important components for the transcription of heterochromatin under certain circumstances. Indeed, heterochromatin is generally methylated and silenced, but it can be transcribed under certain stresses, for example heat stress, without removing silencing marks. Using an elegant forward genetic screen, the authors identified *med14* and *uvh6* mutants in which the reactivation of a silenced transgene by heat stress is compromised. Through whole-genome analysis, they show that mostly transposable elements (TEs) of the Copia and Gypsy families are regulated by MED14, whereas UVH6 has a more narrowed effect. Then, by combining *med14* and *uvh6* with *ddm1* and *mom1* mutations, which allow reactivation of TEs with or without demethylation, they show that MED14 is required for the transcription of methylated TEs. The amount of work is impressive and of very high quality. The conclusions are more advanced for MED14 than UVH6, but studying these two mutants provides a good balance. The manuscript is very well organized and suitable for publication.

We thank the reviewer for her/his very positive comments on our manuscript.

Addressing the following questions would improve the manuscript:

Given that *mom1* has a weaker effect than *ddm1* on the L5 transgene, it would be interesting to look if L5 is more reactivated in the *mom1 med14* double mutant.

As suggested, we have assessed L5 expression in the *mom1 med14* double mutant background using GUS histochemical staining and mining our RNA-seq data. The results show that GUS transcripts from the L5 transgene accumulate at slightly higher levels in *med14-3 mom1-2* relative to *mom1-2*. In addition, although MED14 is required for transcription of the vast majority of TEs depressed in *mom1-2*, our RNA-seq data also indicate that several endogenous TEs are transcriptionally activated in *mom1 med14* double mutants and not necessarily in either single mutants. This suggests that MED14 regulates a layer of silencing at some loci. These new results are presented in the new supplementary figure 12 of the revised manuscript.

Given that, like *mom1*, the *morc* and *main* mutants allow reactivating TEs without affecting methylation, it would be interesting to determine if their effect is also blocked by *med14*.

This is an excellent suggestion that we actually have started addressing by crossing *med14* with *mail1* and *morc6* mutants (*MAIL1* and *MAIN* act in the same silencing pathway, likely as heterodimers; Ikeda et al. 2017). However, due to the necessity of breeding for these new mutant combinations, such analysis would significantly delay the current submission.

Minor comments:

In the abstract, the sentence "MED14 is required for transcription when heterochromatin silencing is destabilized in the absence of stress" is unclear. How heterochromatin silencing is destabilized is explained in the text, but it should be clearly said in abstract.

We have modified the abstract accordingly. The corresponding sentence now reads: "We find that MED14, but not UVH6, is required for transcription when heterochromatin silencing is destabilized in the absence of stress through mutating the MOM1 silencing regulator"

Reviewer #3 (Comments to the Authors (Required)):

Heterochromatin provides a means to tightly pack chromatin and shutdown transcription over relatively large genomic distances. This is particularly important in pericentromeric regions. Despite this large-scale reduction of transcription, as these authors show, heterochromatin can still be activated under stress conditions, despite repressive chromatin marks. In this study the authors have devised a very elegant forward genetic screen to identify key components necessary for transcriptional activation of heterochromatin in detached leaves. The authors go on to clone the underlying genes and characterize them in some detail. The authors show very convincingly that MED14 and UVH6 are key players in this pathway, and likely act together to activate the transcription of many silenced genes in response to heat stress.

This manuscript is extremely well written and very well presented. A number of interesting and well-designed experiments are described and executed and the conclusions drawn are balanced and very well supported. I believe this is a complete study. I see no major gaps or shortcomings. The following are essentially minor suggestions that might help clarify some aspects of the study.

We are grateful to the reviewer for her/his very positive assessment of our work.

(1) The authors exclude H2A.Z, MORC6 and HsfA2 as being major actors in releasing transcription in silenced chromatin. It would be interesting to know if they have considered HsfA1a class transcription factors. These are

upstream of HsfA2 and responsible for the initial response to heat. There are for example several published ChIP-seq datasets for HsfA1 class TFs, so it would be possible to see if these are enriched in the UVH6 and MED14 dependent genes. (Related to this, the authors mention ONSEN, which has HSEs, but it isn't clear if ONSEN is altered in their mutants.

As suggested, we have determined the overlap between HsfA1 binding sites (from Cortijo et al. 2017 Mol. Plant) and all genes upregulated by heat stress, or UVH6 and MED14 dependent genes:

This reveals that only 4-16% of these genes contain HSFA1 binding sites. However, this does not exclude an important role for HSFA1, which could only be clearly examined by analyzing silencing release in heat-stressed *hsfa1* mutants, as we did for H2A.Z, MORC6 and HSFA2. Therefore, we have decided not to include these data in the revised manuscript.

Heat stress-induced ONSEN activation is altered in *med14* and *uvh6* mutants; these data were actually already included in supplementary figure 9. We now mention ONSEN in the main text of the revised manuscript when referring to this figure.

(2) Related to (1) the publicly available DAP-seq data from the Ecker lab might be a useful resource to determine the overall mechanism by which these genes are regulated. Briefly, while heterochromatin is a broad mechanism to shutdown the expression of hundreds or thousands of genes, the specific up-regulation of many of these genes by heat, and the even more specific requirement of a subset of these for UVH6 and MED14 suggests the mechanism is quite specific, suggesting there might be a specific TF responsible. This may well be beyond the scope of this study, but it might be easy to identify from data already available, or it might be interesting if the authors briefly touched on this issue.

We thank the reviewer for this excellent suggestion. We have determined the enrichment in the different TF binding sites identified by the Ecker lab at genes up-regulated by heat stress and genes requiring UVH6 or MED14 for their heat-mediated activation. These data are presented in the new supplementary figure 9 (A, B, C) and commented page 8 of the revised manuscript. Assessing the involvement of the identified TFs in heat stress response will obviously require further studies.

(3) Typos: P3: Notably, how the transcriptional machinery can access to a repressive chromatin environment. And: heading: Transcriptomic analysis of *uhv6* and *med14* mutants in the absence of stress₁₁

This has been corrected in the revised manuscript.

December 4, 2018

RE: Life Science Alliance Manuscript #LSA-2018-00197R

Dr. Olivier Mathieu

Génétique Reproduction et Développement (GReD), CNRS, Inserm, Université Clermont Auvergne

GReD - UMR CNRS 6293, UCA, Inserm 1103

UFR de Médecine, 28 place Henri Dunant

Clermont-Ferrand 63000

France

Dear Dr. Mathieu,

Thank you for submitting your revised manuscript entitled "A role for MED14 and UVH6 in heterochromatin transcription upon destabilization of silencing".

I appreciate the way you responded to the reviewers' concerns and the additional data provided, and we would be happy to publish your paper in Life Science Alliance pending minor revision. I would thus like to invite you to submit a final version of your manuscript, in which you down-tone in both abstract and manuscript text the possibility that MED14 requires wild-type epigenetic marks for its function. While possible, further proof would be needed (as you also outline yourself) to conclude this.

A. FINAL FILES:

-- High-resolution figure, supplementary figure and video files uploaded as individual files: See our detailed guidelines for preparing your production-ready images, <http://life-science-alliance.org/authorguide>

B. MANUSCRIPT ORGANIZATION AND FORMATTING:

Full guidelines are available on our Instructions for Authors page, <http://life-science-alliance.org/authorguide>

Sincerely,

December 5, 2018

RE: Life Science Alliance Manuscript #LSA-2018-00197RR

Dr. Olivier Mathieu

Génétique Reproduction et Développement (GReD), CNRS, Inserm, Université Clermont Auvergne

GReD - UMR CNRS 6293, UCA, Inserm 1103

UFR de Médecine, 28 place Henri Dunant

Clermont-Ferrand 63000

France

Dear Dr. Mathieu,

Thank you for submitting your Research Article entitled "A role for MED14 and UVH6 in heterochromatin transcription upon destabilization of silencing". I appreciate the introduced changes, and it is a pleasure to let you know that your manuscript is now accepted for publication in Life Science Alliance. Congratulations on this interesting work.

DISTRIBUTION OF MATERIALS:

Again, congratulations on a very nice paper. I hope you found the review process to be constructive and are pleased with how the manuscript was handled editorially. We look forward to future exciting submissions from your lab.

Sincerely,

Andrea Leibfried, PhD
Executive Editor
Life Science Alliance
Meyershofstr. 1
69117 Heidelberg, Germany
t +49 6221 8891 502
e a.leibfried@life-science-alliance.org
www.life-science-alliance.org